# What Drives Compositional Generalization in Visual Generative Models?

## Abstract

Compositional generalization, the ability to generate novel combinations of known concepts, is a key ingredient for visual generative models. Yet, not all mechanisms that enable or inhibit it are fully understood. In this work, we conduct a systematic study of how various design choices influence compositional generalization in image and video generation in a positive or negative way. Through controlled experiments, we identify two key factors: (i) whether the training objective operates on a discrete or continuous distribution, and (ii) to what extent conditioning provides information about the constituent concepts during training. Building on these insights, we show that relaxing the MaskGIT discrete loss with an auxiliary continuous JEPA-based objective can improve compositional performance in discrete models like MaskGIT.

## 1 Introduction

Visual generative models, such as diffusion models (Kingma et al., 2021; Nichol & Dhariwal, 2021; Yang et al., 2023) and generative transformers (Wang et al., 2022; Kim et al., 2023; Hudson & Zitnick, 2021) can generate high-fidelity images and videos (Villegas et al., 2022; Ho et al., 2022; Karras et al., 2020), especially when trained on large amounts of data. However, are these models able to achieve robust *compositional generalization* or are they simply "interpolating from their training data?" That is, can they systematically decompose the observed data into its underlying causal factors or "concepts" (e.g., objects, attributes) and synthesize novel combinations not seen during training (Zhao et al., 2022; Wiedemer et al., 2023; Favero et al., 2025)?

Despite significant progress in generative modeling, compositional generalization in current models remains inconsistent. While some studies report successes on targeted compositional tasks (Wiedemer et al., 2025; Gaudi et al., 2025), others reveal notable limitations, particularly when trying to generate complex scenes or novel combinations of known elements (An et al., 2023; Keysers et al., 2019). A representative example is shown in Figure 1: when we trained two types of state-of-the-art generative models—DiT (Peebles & Xie, 2023) and MaskGIT (Chang et al., 2022)—on images of, say, "non-smiling, women with blonde hair" and "smiling men with black hair", and then conditioned them to generate a "non-smiling man with blonde hair"—a novel combination of known factors—one model can do so (DiT) while the other struggles to do so (MaskGIT). This contrast raises a central question:

*What are factors that enable or hamper compositional generalization in visual generative models?*

In this work, we dissect modern visual generative models into three key components: (i) *Tokenizer*, which defines the representation space; (ii) *Generative model*, which generates samples in the tokenizer defined space guided by the conditioning signal, and (iii) *Conditioning signal*, which specifies the compositional factors to be generated. This dissection lays the ground for the following central research questions:

- **RQ1**: *Does the type of the tokenizer affect compositional generalization?* (VAE with KL regularization vs. VQ-VAE with quantization/commitment regularization.)
- **RQ2**: *How does the generative model design affect compositionality?* In particular, does it matter whether the modeled distribution is continuous or discrete (e.g., continuous latents vs. discrete tokens)? And is a denoising-based objective essential, or does a masking-based loss suffice?

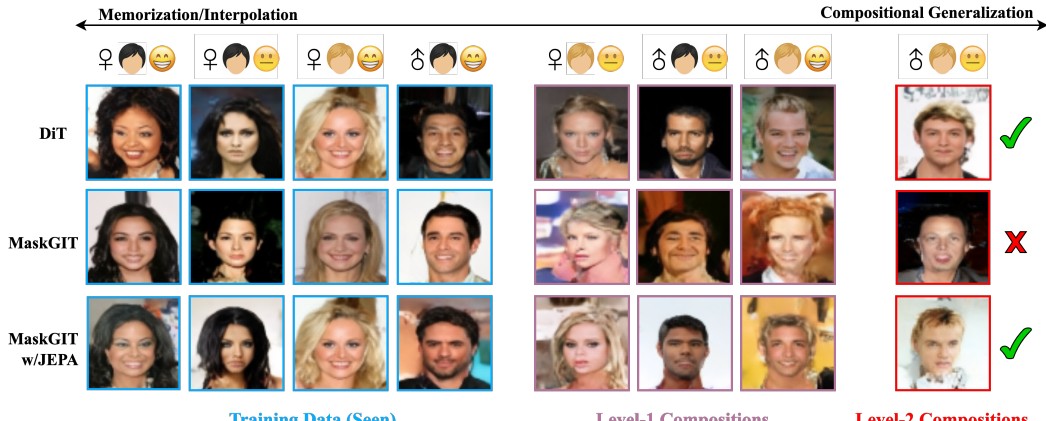

Figure 1: **Compositional Generalization Analysis.** We evaluate how generative models (MaskGIT, DiT) generalize to novel compositions of three binary factors on CelebA: gender, hair color, and smile. Models are trained on four combinations (blue) and evaluated on two sets of novel compositions (pink: Level-1 (one-factor change), red: Level-2 (two-factor change)). While MaskGIT (2nd row) shows poor compositional generalization, DiT (1st row) exhibits better compositional generalization. We also show that we can improve MaskGIT's compositional generalization abilities by augmenting its training objective with a JEPA-based training objective (3rd row).

- **RQ3**: *How does the nature of conditioning during training influence compositionality?* Must conditioning have the exact factors of the data-generating process, or can they rely on quantized or missing abstractions (e.g., "red" instead of a precise RGB value, or hiding "smiling" in {"blond", "smiling", "girl"})?

- **RQ4**: Guided by our findings from the previous questions, *can we intervene on non-compositional models to endow them with compositional capabilities?*

To shed light on these questions, we conduct controlled experiments across a range of architectural and training design choices. The results consistently indicate that models trained to learn a *continuous distribution*, by their training objective, exhibit stronger compositional abilities than models trained to model a categorical distribution. Furthermore, we find that *providing full conditioning information of the generating factors during training* is critical; quantized or partial conditioning leads to weaker compositional generalization. On the other hand, training the tokenizer with a quantization bottleneck has no significant effect on the downstream compositional generalization.

Guided by our findings and to further examine why discrete modeling hinders compositional generalization, we augment MaskGIT's discrete, categorical training objective with a Joint Embedding Predictive Architecture (JEPA) objective (LeCun, 2022). This modification introduces continuous latent targets and yields clear improvements in MaskGIT's compositional performance. At the same time, JEPA-trained models exhibit more disentangled intermediate representations, suggesting that predictive continuous objectives can shape the internal structure to retain compositionality in discrete generative models.

> **Summary of contributions**: Our study shows that achieving robust compositional generalization in visual generative models can be facilitated by continuous distribution modeling and full, non-quantized conditioning signals. In addition, we show that continuous representation learning objectives such as JEPA can further enhance compositional generalization in discrete models like MaskGIT.

## 2 RELATED WORKS

Factorization and compositional generalization have been extensively studied across a range of architectures (Wu & Goodman, 2019; Li et al., 2024a; Yang et al., 2024b), as they are central to building models that can systematically capture and recombine the underlying factors of variation in data. Prior work has approached this challenge from multiple angles. On the theoretical side, several studies seek first-principles characterizations of compositionality based on the data-generating process (Wiedemer et al., 2023) or provide provable guarantees through structural constraints such as object-centric autoencoders (Wiedemer et al., 2024). Our work differs by adopting a practical, empirical perspective, systematically probing how architectural and training choices shape compositional generalization in visual generative models.

Complementary to these theoretical accounts, recent empirical investigations highlight how large-scale models such as CLIP (Radford et al., 2021) exhibit degrees of compositionality, often tied to the properties of their training data (Kempf et al., 2025; Wiedemer et al., 2025). Our focus in this work is not on data, but the importance of different architectural, training-specific, or conditioning choices that enhance compositional abilites.

Within generative modeling specifically, Okawa et al. (2024) studied compositional generalization solely for diffusion models through controlled experiments. We extend this line of inquiry along two dimensions: (i) by considering a broader set of generative frameworks, including both continuous and discrete models, and (ii) by examining additional data modalities such as video. Our work also connects to the rich literature on disentangled representation learning (Higgins et al., 2018; Caselles-Dupré et al., 2019), which aims to align individual latent units with interpretable factors of variation. While disentanglement work focuses on representation quality, our study emphasizes the downstream generative consequences: we analyze how different training objectives and conditioning strategies enable—or hinder—the synthesis of novel compositions.

## 3 EXPERIMENTAL SETUP

Our goal is to identify which architectural and training choices enable or hinder compositional generalization (as seen in Figure 1). To this end, we design controlled comparisons that systematically vary key factors-representation space, training objective, and conditioning information levels-while holding others fixed. This allows us to "interpolate" between existing models such as DiT and MaskGIT and to isolate the contributions of individual design decisions.

**"Interpolation" between DiT and MaskGIT**   The main differences between DiT and MaskGIT lie in the following design choices: the tokenizer (VAE/VQ-VAE for DiT vs. strictly VQ-VAE for MaskGIT), the training objective (masking-based for MaskGIT, absent in DiT), the loss function (diffusion in DiT vs. categorical negative log-likelihood (NLL) in MaskGIT), and the nature of the latent output distribution (continuous for DiT vs. categorical discrete for MaskGIT). Our goal is to change these choices one at a time to identify their effect on compositional generalization.

Since the initial publications of DiT and MaskGIT, subsequent work has effectively implemented some of these interpolations; see Table 1. For example, MAR (Li et al., 2024b) differs from DiT only in adopting a masking-based prediction while retaining the per-token diffusion objective. Similarly, GIVT (Tschannen et al., 2024) replaces replaces MaskGIT's standard categorical NLL with a Gaussian mixture negative log-likelihood (GMM-NLL), where the token distribution is modeled as a mixture of Gaussians (Reynolds et al., 2009) (which allows GIVT to operate in a continuous space without employing the denoising diffusion objective). We thus adopt these established variations as the interpolations in our experiments. Further architectural and training details for all models are provided in Section A.

**Testing compositional generalization**   We train models on pairs of images conditioned on tuples specifying the underlying factors (e.g., object color). We train only on a subset of the possible factor combinations and evaluate compositional generalization on held-out combinations. We distinguish between *level-1* compositions, which differ from the nearest combination seen during training by one factor, and *level-2* compositions, which differ by two factors. Following Okawa et al. (2024), we focus on three binary independent factors (unless stated otherwise) in simple synthetic images in the

Table 1: **Summary of generative models.**

| Model | Masking-based | Training loss | Latent output distribution |
|---|---|---|---|
| MaskGIT | ✓ | Categorical NLL | discrete |
| GIVT | ✓ | GMM-NLL | continuous |
| MAR | ✓ | Diffusion | continuous |
| DiT | ✗ | Diffusion | continuous |

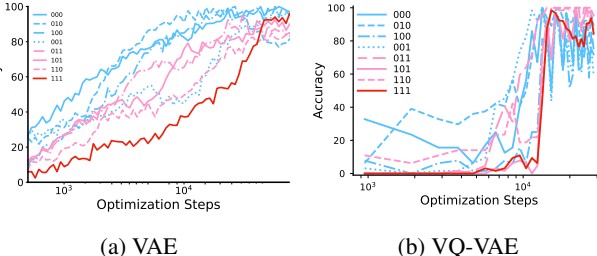

(a) VAE  (b) VQ-VAE

Figure 2: **DiT exhibits compositional generalization regardless of the type of tokenizer used.** While the training dynamics differ, DiT shows compositional generalization at end of training. The blue, pink, and red curves show linear probe accuracies for the training data, level-1 compositions, or level-2 compositions, respectively. Consistent results are observed for MAR (Figure 7) and across video datasets (see Figure 8).

main text (Shapes2D). In addition, we also verify the validity of our findings across spaces with multi-valued factors (Shapes3D), real-world images (CelebA), and also to videos (CLEVRER-Kubric). More information for each dataset is provided in Appendix B.

**Evaluation** To evaluate whether generated images faithfully capture the intended factors, we train probes for each factor to classify its presence in the generated outputs. We train the probes on *all* possible combinations, including those held-out from generative model training. Following (Okawa et al., 2024; Park et al., 2024), a composition is considered correct if all factors' probe probability is at least 0.5. For visualizing training dynamics, we use blue curves for compositions seen in training, pink curves for held-out level-1 compositions, and red curves for held-out level-2 compositions. Performance on level-2 compositions serves as a particularly informative metric for compositional generalization, as they require the model to generate multiple novel factors simultaneously.

## 4 DESIGN CHOICES THAT DRIVE COMPOSITIONALITY

In this section, we leverage the setup introduced in Section 3 to systematically rule out factors that are either irrelevant for DiT's compositional generalization or not present in MaskGIT. This process isolates the design elements that are essential for enabling compositional generalization in DiT but not MaskGIT. We focus on results from the Shapes2D dataset in the following. However, our findings also extend to more complex, real-world, and video datasets (see Section F).

**Does the choice of tokenizer matter?** The tokenizer is one of the main differences between DiT and MaskGIT (Table 1). To isolate and assess its effects on compositionality (**RQ1**), we evaluate the same second-stage architecture (DiT) with two different tokenizers: a discrete tokenizer (VQ-VAE) and a continuous tokenizer (VAE). Additional implementation details are provided in Section C.1.

Figure 2 shows that DiT achieves comparable compositional generalization performance across both tokenizer types by the end of training. The training dynamics, however, differ: with a continuous tokenizer progress is more gradual and steady, whereas with a discrete tokenizer, compositional generalization emerges more abruptly. Further, we find that the discrete tokenizer is more sensitive to learning rates.[1] These results suggest that the choice of tokenizer regularization, *vector quantization*

---

[1] A detailed analysis of these differences is left for future work.

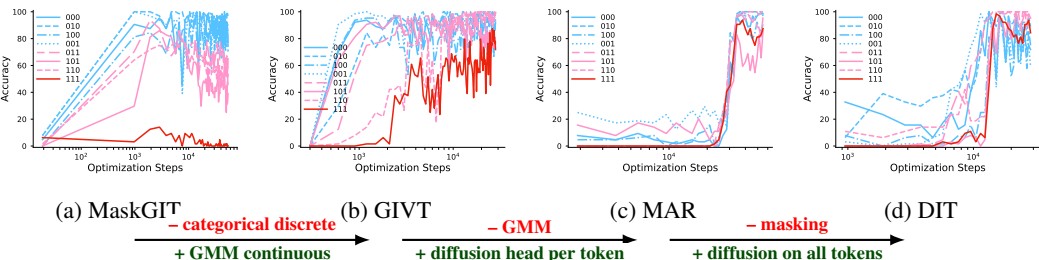

Figure 3: **Compositional generalization performance on Shapes2D across different model architectures.** Models that learn continuous distributions (DiT, MAR, and GIVT) consistently show better level-2 compositions than MaskGIT, with the decisive shift in performance occurring at the categorical-to-continuous intervention. The blue, pink, and red curves denote training, level-1, and level-2 compositions, respectively. Consistent results are observed for CLEVRER-Kubric (Figure 9).

(VQ-VAE) vs. *KL* (VAE), does not erode compositionality and is unlikely to be the critical factor explaining why DiT exhibits compositional generalization while MaskGIT does not.

> **Finding 1:** The choice of tokenizer does not fundamentally alter the compositional abilities of a generative model. It primarily affects training efficiency and stability.

Since tokenizer choice does not affect compositionality and MaskGIT requires discrete tokens, we fix the tokenizer to VQ-VAE to avoid *VQ* vs. *KL* regularization confounds. Discrete models use codebook indices; continuous models use pre-quantization latents. See Section C.1 for details.

**Does the masking matter?**    Another main difference between DiT and MaskGIT is their objective: MaskGIT employs a MaskGIT uses a masking-based loss[2], whereas DiT relies on the standard denoising approach (Table 1). MAR can be viewed as a hybrid or "interpolation" between these two: it combines a masking-based objective with a diffusion loss per token.

Figure 3c shows that MAR achieves strong compositional generalization by end of training. This indicates that the presence or absence of masking in the training objective does not critically impact the model's ability to generalize compositionally.

**Is it the diffusion loss?**    We look at another major difference between DiT and MAR vs MaskGIT: their training losses. GIVT (Tschannen et al., 2024) provides a bridge between MAR and MaskGIT on the objective axis by replacing MaskGIT's categorical head with a continuous Gaussian Mixture Model (GMM). Note that this intervention can be viewed as replacing the diffusion loss of MAR and DiT with maximum likelihood training. This is done by using the continuous parameters of a GMM (Reynolds et al., 2009) while preserving the continuous output space. Figure 3a → 3b shows that the MaskGIT → GIVT intervention (categorical → continuous) yields a significant performance jump in compositional generalization, whereas GIVT → MAR (GMM → diffusion) brings comparatively smaller gains as seen in Figure 3b →3c. This shows that the continuous training objective on a continuous output space, rather than the exact form of the objective (denoising vs. parameter prediction), is the key factor for enabling compositional generalization.

**Output space continuity is the key**    Through systematically controlling for the main differences between DiT and MaskGIT, including the choice of the tokenizer, masking strategies and loss functions, we find that none of these factors critically impact compositional generalization. The remaining distinguishing factor is the nature of the predicted outputs: DiT predicts continuous-valued quantities, while MaskGIT predicts discrete tokens. Based on our experiments, we can conclude that this difference in output representation (and thereby the respective objective) is the key factor underlying DiT's ability to achieve compositional generalization, not observed in MaskGIT.

---

[2]"Masking-based" denotes predicting a subset of tokens conditioned on the remainder, via continuous vector masks or discrete code masks.

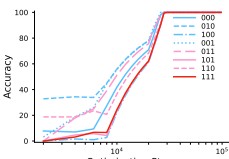 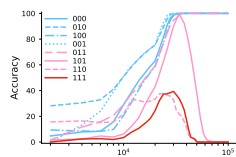 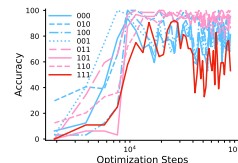 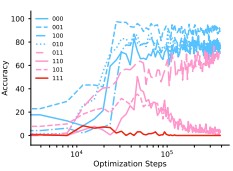

(a) Full-information conditioning

(b) Full-information conditioning + dropout

(c) Quantized conditioning

(d) Quantized conditioning + dropout

Figure 4: **Comparison of conditioning information levels and their impact on compositional generalization in DiT on Shapes2D.** **(a)** Continuous (full-information) conditioning leads to uniform convergence across all compositions. **(b)** Label dropout conditioning leads to inconsistent generalization; several unseen compositions fail completely. **(c)** Discrete (quantized) conditioning leads to partial generalization, with some failing samples. **(d)** Discrete (quantized) conditioning with dropout, the most severe loss of information, leads to the most failure. Shaded areas indicate standard deviation across three different seeds. We provide additional results in Section C.3. The blue curves show performance on training data, pink curves depict level-1 compositions, and red curve denotes level-2 compositions.

> **Finding 2:** Generative models trained to model continuous distributions reliably exhibit strong compositionality, whereas models trained on discrete, categorical distributions do not.

## 5    IMPORTANCE OF THE CONDITIONING INFORMATION LEVELS

After identifying the critical design choices in both stages of the generative model, we investigate how different forms of conditioning affect compositional generalization (**RQ3**). In prior work on compositionality in diffusion models, Okawa et al. (2024) conditioned models on complete, continuous factor representations (e.g., raw hue or size). In more realistic scenarios, however, factors are often quantized (e.g., the word "red" instead of the exact hue) or provided as lossy description (i.e., some factors are missing in the conditioning signal). For quantized signals, we convert continuous signals into discrete binary signals. For lossy signals, we randomly drop each factor with a 10% probability, so that, on average, one in ten factors is missing.

Figure 4c shows that compositional generalization becomes less stable under quantized conditioning. When some factors present in the image are occasionally missing from the conditioning signal, compositional generalization typically fails (Figure 4b). Combining both conditions–a setting common in practice–produces the strongest negative effect (Figure 4d). Overall, these results show that limited information–whether through quantization or incomplete conditioning– can impair compositional generalization, even when all factors are provided during generation.

> **Finding 3:** Full, precise conditioning is critical for robust compositional generalization. Models trained with quantized or incomplete signals show poor or inconsistent recombination of factors.

## 6    ENHANCING COMPOSITIONAL LEARNING WITH JOINT EMBEDDING PREDICTIVE ARCHITECTURES

In the previous section, we identified a key factor that can hinder compositional generalization in modern generative models: training objectives that operate over discrete, categorical distributions. Despite this, discrete training objectives–such as the one used in MaskGIT–offer advantages in, e.g., sampling speed compared to alternatives like DiT. This raises an important question: Can we retain these advantages while also improving compositional generalization?

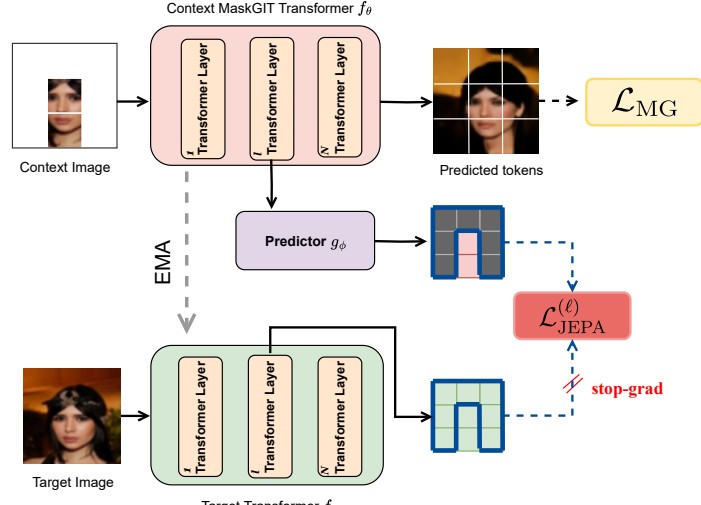

Figure 5: **An overview of MaskGIT combined with the JEPA-based training objective.** We apply the JEPA loss at specific layers ($l$) on an intermediate masked token representation in the transformer $(H_C^{(l)})$ and train a lightweight predictor to reconstruct target states $(H_T^{(l)})$ using MSE as an error metric and a stop-gradient signal to avoid representation collapse.

## 6.1 An Auxiliary Objective for MaskGIT

Building on our previous findings, we extend MaskGIT's training objective with an auxiliary continuous representational loss, similar to Joint Embedding Predictive Architecture (JEPA) (LeCun, 2022; Bardes et al., 2024). JEPA learns to reconstruct target patch representations by using (other) context patches from the same image. This auxiliary objective resembles MaskGIT's own masking-based formulation but operates directly on continuous latent representations rather than discrete tokens.

Formally, following MaskGIT, we encode the input image or video $x$ into the discrete tokens $z = \{1 \dots K\}^N$ using a VQ-VAE. MaskGIT is then trained to reconstruct masked subset of tokens $z_\mathcal{M}$ from the unmasked tokens $z_{\bar{\mathcal{M}}}$ ($\mathcal{L}_\text{MG}$). For images, masking is applied spatially, whereas for videos, it is applied causally. Now, we add the auxiliary representation alignment objective based on JEPA ($\mathcal{L}_\text{JEPA}$). Given the context latent representations $H_C^{(l)}$ from selected intermediate layers $l$, the model is trained to reconstruct the target latent representations $H_T^{(l)}$. We use Mean Squared Error (MSE) with a stop-gradient on the target representations from an EMA version of the model to stabilize training. In the final objective, we jointly optimize $\mathcal{L}_\text{MG}$ and $\mathcal{L}_\text{JEPA}$. More information about the objective and results on video generation tasks can be found in Section D of the Appendix.

While this auxiliary loss resembles the recent REPresentation Alignment (REPA) framework (Yu et al., 2025), REPA aligns generative latents with an external pretrained encoder. In contrast, our JEPA-based objective is completely self-supervised, structuring the model's own intermediate representations with a continuous loss to support our findings and, in turn, compositional generalization.

## 6.2 Results and Analysis

Extending MaskGIT with the auxiliary objective improves compositional generalization on images (Figures 1 and 6b), particularly for level-2 compositions. In contrast, without the auxiliary loss, MaskGIT exhibits little to no compositional generalization (Figure 6a). These results reinforce our earlier finding that having an objective operating on continuous outputs (or, by extension, intermediate latent representations) is critical for enabling compositional generalization.

To better understand the effect of our auxiliary loss, we analyze its effect on the learned representations below. We apply techniques from mechanistic interpretability (Bereska & Gavves, 2024), which allow us to probe how individual components encode specific factors and interact during generation.

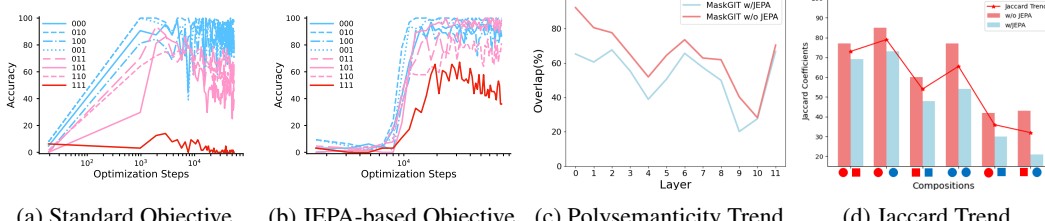

(a) Standard Objective  (b) JEPA-based Objective  (c) Polysemanticity Trend  (d) Jaccard Trend

Figure 6: Comparison of linear probe accuracy for MaskGIT with Standard (a) and JEPA-based (b) training objectives for Shapes2D. The JEPA-based training objective clearly enhances compositional abilities even though it cannot fully compensate the problems introduced by the discrete space. We also see lower polysemanticty between attention heads (c) and a decreasing Jaccard trend over shared circuits (d). The blue curves show performance on training data, pink curves depict level-1 compositions, and red curve denotes level-2 compositions. Consistent results for videos shown in Figure 12.

**Polysemanticity in attention heads** An attention head can easily become *polysemantic*, meaning that it "attends to" multiple distinct and unrelated features simultaneously. This can occur due to phenomena like superposition (Elhage et al., 2022). However, superposition, resulting in polysemanticity, causes undesired entanglement: if an attention head responsible for processing `color` also strongly activates for `shapes`, it becomes difficult for the model to disentangle these factors.

To quantify polysemanticity, we need to assess the causal effect of each attention head on the predefined visual factors/features. We pass pairs of images that differ only in a single factor (e.g., a *red* large square vs. *blue* large square) and compute the feature similarity for the global token. We then systematically deactivate attention heads and re-compute the similarity. A head is considered polysemantic if the feature similarity difference exceeds a threshold, suggesting that it attends to multiple, entangled factors that may hinder compositional control.

Figure 6c shows that fewer attention heads are polysemantic in MaskGIT with the auxiliary continuous loss. This suggests that the factors are less entangled in the generative model's representations.

**Mechanistic similarity using circuits** The above analysis shows that attention heads mix up some features, but it only provides a coarse view of representational entanglement. To probe deeper into how concepts are internally represented and transformed across layers, we analyze the model's *internal circuitry* (Olah et al., 2020)-groups of neurons that jointly implement specific mechanisms.

We aim to identify the most influential components that generate each composition. To do so, we focus on MLP neurons within each transformer block, since prior work has shown that these often encode semantic factors (Geva et al., 2020; Dai et al., 2021). For each composition, we identify the top-$k$ most influential neurons per layer using activation patching (Vig et al., 2020; Meng et al., 2022). Specifically, we compute each neuron's *indirect effect* (Pearl, 2013), which captures its downstream influence on the model's output rather than just its direct contribution. This is achieved by comparing the output when running the model on a factual input (the correct composition condition, e.g., "red circle") with the output when running on a counterfactual input (the same condition but with a key intermediate activation corrupted, e.g., replaced with noise), and then patching in the neuron's clean activation. Intuitively, this measures how strongly a neuron shapes the generated image or video.

We then compare the overlap of the top-$k$ neurons across different compositions. For this, we use the Jaccard index, following Kempf et al. (2025). A high overlap implies that the same neuron is used for multiple factors, suggesting that circuits are more entangled. In contrast, a low overlap suggests more separate circuits. We provide more information in Section D.5.

Figure 6d shows that adding the auxiliary JEPA objective consistently reduces neuron overlap across layers, particularly for composition pairs where both factors differ (e.g., `red circle-blue square`). This suggests that the model has learned more factor-specific circuits.

> **Finding 4:** A JEPA-based training objective induces more disentangled and semantically structured representations, and enables stronger compositional generalization.

## 7 DISCUSSION

**Do our results extend to world models in the real domain?** A common critique of compositional generalization works in generative models is their reliance on toy datasets. To validate our conclusions on real-world scale, we curate compositional splits on a driving scenes video dataset, CoVLA (Arai et al., 2025): factors are *time of day* (day $\odot$/night $\bigcirc$) and *turn direction* (left/straight/right), and we hold out $\odot\rightarrow$ and $\bigcirc\leftarrow$. We instantiate the current SoTA lightweight driving world model, *Orbis* (Mousakhan et al., 2025), in two variants with matched training and inference budgets: Orbis-DiT (continuous) and Orbis-MaskGIT (categorical). We evaluate compositionality using a Compositional Retrieval Accuracy (CRA) metric: the fraction of nearest neighbors in a V-JEPA2 embedding (Assran et al., 2025; Luo et al., 2024) that share the conditioning factors (see App. G). In Tab. 6, DiT substantially outperforms MaskGIT on the *novel* compositions: on $\odot\rightarrow$, DiT can faithfully generate 0.47 vs. 0.18 (absolute **+29.7 pp**, relative **+161 %**); on $\bigcirc\leftarrow$, DiT reaches 0.43 vs. 0.14 (absolute **+29.0 pp**, relative **+207 %**). Per-split nearest-neighbor ratios further indicate that MaskGIT tends to collapse toward seen combinations, whereas DiT allocates more mass to the intended novel target (qualitative examples in Fig. 19). While absolute accuracies are modest—reflecting the difficulty of real-world video with implicit factors in context frames—the trend aligns with our controlled interventions in Finding 2, supporting that *continuous objectives improve compositionality under complete conditioning* relative to categorical objectives. A broader study of real-world challenges, implicit conditioning via text/images, and scaling is left to future work.

**Do our results extend to language?** Although our study focuses on *visual* generative models, compositional generalization has been a central theme in language modeling as well (Yang et al., 2024a; Furuta et al., 2023; Sakai et al., 2025). Our findings suggest that a continuous objective helps compositional generalization. However, does this also transfer to the language modality? To answer this, we study compositional generalization for Llama-3.2 (Dubey et al., 2024) on the Points24 dataset (Chu et al., 2025). In this task, the model is given four cards and a target value, and must construct an arithmetic expression that uses each card exactly once to reach the specified target number. Training involves single rules (restricting the target value or doubling the value of red-suit cards), while the (compositional) test split requires composing both rules simultaneously (details in Section H).

We compare two reasoning mechanisms: standard Chain-of-Thought (CoT) (Wei et al., 2023) and its continuous variant, COntinuous-Chain-Of-Thought (COCONUT) (Hao et al., 2024). Here, reasoning mechanisms play a role analogous to training objectives in our previous experiments: shaping whether intermediate steps are treated as discrete symbolic traces (CoT) or continuous thoughts (COCONUT). We find that COCONUT yields higher accuracy (12.39%) than CoT (4.82%) on the compositional splits, suggesting that the advantages of continuous objectives observed earlier may carry over to language. We leave further investigation for future work.

## 8 CONCLUSION

Our work aims to answer a fundamental question for reliable generative modeling- *what drives compositional generalization in visual generative models?* Through a series of controlled experiments across architectures, objectives, and conditioning information, we identified two consistent principles. First, models trained to represent continuous distributions exhibit strong compositional generalization, while discrete categorical objectives inhibit it. Second, complete conditioning is essential- quantized or incomplete conditioning leads to unstable or failed compositions. Given these insights, we introduced a JEPA-based auxiliary loss that improves compositionality for discrete models, and our mechanistic analysis suggests that it promotes learning disentangled, factor-specific circuits.

These findings provide a foundation for designing objectives and conditioning strategies that better preserve semantic structure and enable novel compositions beyond the training distribution. We see this as a step towards building causal generative models that can move beyond memorization, achieving systematic and reliable generalization needed for truly creative and trustworthy AI systems.

## THE USE OF LARGE LANGUAGE MODELS (LLMS)

We used GitHub Copilot (2025) as a coding assistant during implementation and GPT-5 (OpenAI, 2025) to polish the writing. All core contributions and the initial draft were done by the authors.

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

## A  MODELS

We selected a representative set of generative models to span key design axes relevant to compositional generalization. **Diffusion Transformers (DiTs)** (Peebles & Xie, 2023) operate in a continuous latent space derived from a VAE and are trained with a standard denoising objective; conditioning is applied via adaptive layer normalization. In contrast, **MaskGIT** (Chang et al., 2022) works in a discrete latent space obtained from a VQ-VAE and employs a bidirectional transformer to iteratively predict masked tokens, corresponding to a categorical, masking-based objective. **MAR** (Li et al., 2024b) also uses a continuous latent space but masks subsets of latent tokens and leverages an encoder-decoder to generate intermediate conditions for a diffusion head that denoises each masked token independently, combining aspects of masking and denoising. Together, these models cover continuous versus discrete representations, denoising versus masking objectives, and a variety of conditioning strategies.

### A.1  DIFFUSION TRANSFORMER

To evaluate models in continuous representation spaces, we train **Diffusion Transformers** (DiTs) Peebles & Xie (2023) which adapt the Transformer architecture Vaswani et al. (2017) within the Latent Diffusion Model framework Nichol & Dhariwal (2021). Initially, a pretrained VAE encodes an input image $x$ into a compressed latent representation $z = E(x)$ which is processed into patch embeddings $s$. A standard forward diffusion process then progressively adds Gaussian noise to this latent representation $s$ over $t$ timesteps. The core of the model is a Transformer network trained to denoise the noisy latent $s_t$ by predicting the added noise $\epsilon_t$ at each timestep $t$, typically minimizing $L = \mathbb{E}_{s_0,\epsilon,t,c}\left[\|\epsilon - \epsilon_\theta(s_t, t, c)\|_2^2\right]$, where $\mathbb{E}$ denotes the expectation over the initial latent representation, $\epsilon_\theta$ is the noise predicted by the Transformer network with parameters $\theta$, and $\|\circ\|_2^2$ denotes the squared L2 norm (Euclidean norm). We implement conditional generation by integrating guidance information $c$ into the Transformer blocks using adaptive layer normalization (adaLN), which modulates network activations based on $c$ and $t$.

### A.2  MAR

**MAR** Li et al. (2024b) explores generation within a continuous latent space defined by a VAE while preserving autoregressive principles- iteratively generating a set of tokens conditioned on previously generated tokens. It consists of an encoder, a decoder, and a diffusion head, with both encoder and decoder consisting of a stack of self-attention blocks. Specifically, MAR employs a pretrained VAE to encode an image into a latent representation $\mathcal{Z} \in \mathbb{R}^{h \times w \times d}$ while randomly masking a subset of image tokens within $\mathcal{Z}$. Let $\mathcal{U} = \{z_{i_{\setminus M}}\}_{i=1}^{N_{\setminus M}}$ and $\mathcal{M} = \{z_i\}_{i=1}^{N}$ represent the set of unmasked and masked tokens, respectively, where each $z_{\setminus M}$ and $z$ corresponds to a spatial element in $Z$. Here, $N_{\setminus M}$ denotes the number of unmasked tokens, while $N$ represents the number of masked tokens. The encoder processes $\mathcal{U}$ to extract latent features. The decoder then takes as input both these latent features and a set of learnable tokens of size $N$, each corresponding to a masked token in $\mathcal{M}$. It subsequently generates the features of these learnable tokens, referred to as conditions. Finally, the diffusion head independently performs a denoising process on each masked token $z_i$ using its corresponding condition.

### A.3  MASKGIT

**MaskGIT** or Masked Generative Image Transformer Chang et al. (2022) is an autoregressive model operating in a discrete latent space. An input data sample $x$ is first encoded into a sequence of discrete visual tokens $E_{VQ}(x) \in \{1, ..., K\}^N$ where $E_{VQ}$ is the encoder trained with a VQ-VAE objective and $N$ is the sequence length. The standard MaskGIT objective ($\mathcal{L}_{MG}$) focuses on reconstructing masked input tokens. We employ block-masking Bardes et al. (2024) for images and causal block masking for videos Bardes et al. (2024) to sample a mask $\mathcal{M} \subset \{1, ..., N\}$. The MaskGIT Transformer $T$ predicts the probability distribution $p(z_i|z_{\setminus M})$ for each masked token $z_i$ ($i \in M$) based on unmasked token $z_{\setminus M}$. The loss is the negative log-likelihood of the true tokens at the masked positions $\mathcal{L}_{MG} = -\mathbb{E}_{x,M}\left[\sum_{i \in M} \log p(z_i|z_{\setminus M})\right]$.

MaskGIT then learns a bidirectional transformer model $p_\theta(z|c)$ trained via the masking procedure, predicting masked tokens based on unmasked one and conditioning $c$. Generation is performed

iteratively, starting with all tokens masked and progressively predicting and committing to high-confidence tokens. Conditioning $c$ is embedded to the token sequence using a two-layer MLP and integrated using adaptive layer normalization (adaLN).

# B  DATASETS

Below, we provide an overview of the general characteristics and qualitative examples of the three datasets used in our paper. Results for generation from the generative models can be found in Section F.

## B.1  SHAPES2D

Shapes2D is a synthetic dataset introduced by Okawa et al. Okawa et al. (2024) to systematically study compositional generalization in diffusion models. Each image is composed of a single geometric object characterized by three binary-valued concepts: *shape*, *color*, and *size*, with the composition space defined as `shape={circle, triangle}`, `color={red, blue}`, `size={large, small}`. Unless stated otherwise, these concepts are encoded as binary tuples—e.g., the tuple `000` represents a large, red circle. This structured format allows the use of concept tuples as conditioning signals during diffusion model training, mimicking abstract textual prompts. The dataset is constructed by uniformly sampling over the concept combinations, yielding 1500 images per training concept class and a total of 6000 images. Although the concept variables are nominally discrete, the *size* and *color* attributes are realized through a spectrum of sizes within the big and small labels and hues within the red and blue labels, resulting in tightly clustered yet visually diverse instantiations. These clusters are sparsely distributed and exhibit limited continuity in the pixel space, effectively forming "islands" within the data manifold. Such fragmentation introduces discontinuities in the data-generating factors space that further complicate the model's ability to learn smooth conceptual compositional interpolations.

## B.2  SHAPES3D

Shapes3D (Burgess & Kim, 2018) is a synthetic dataset of rendered 3D scenes designed to study disentanglement and compositional generalization. Each image depicts a single object in a colored room, generated by varying six independent factors: floor color (10 values), wall color (10), object color (10), object shape (4), object size (2), and camera azimuth (15). The full dataset contains 480,000 images, corresponding to all possible combinations of these factors. This makes it a significantly more complex data space than Shapes2D and its fully factorial structure makes Shapes3D well-suited for systematic generalization experiments.

In our experiments, we construct compositional splits by selecting three of the six factors: object color, object shape, and object size. These factors define the conditioning space used for both model training and evaluation. The total number of possible compositions from these three factors is 240.

The model is conditioned on the specific colour, shape, and size values. For compositional evaluation, we group the 240 compositions into 8 supergroups to structure the train–test split and to order results by degree of novelty. As in Shapes2D, we define three levels of novelty: first level includes compositions seen during training. second and third levels are used for testing and differ from the training set in one and two factors, respectively, with level three representing the highest degree of novelty.

We partition the factors of color, shape, and size into two supergroups based on predefined thresholds. Specifically, for the color factor, which consists of ten categories, Supergroup 0 includes the first seven colors (red, orange, yellow, green, cyan, teal, blue), and Supergroup 1 includes the remaining three (indigo, purple, pink). For shape, which contains three categories, Supergroup 0 includes cube and cylinder, and Supergroup 1 includes sphere. Finally, for size, which is defined over ordinal values from 0 to 8, values 0 through 5 belong to Supergroup 0, while values 6 and 7 belong to Supergroup 1.

### B.3 CelebA

To assess the robustness of our findings under real-world settings, we extend our evaluation to the CelebA dataset. Following prior work, we select three visually distinguishable attributes as concept variables: *Gender*, *Smiling*, and *Hair Color*, with the composition space defined as Gender={Male, Female}, Smiling={Smiling, Not Smiling}, Hair Color={Black Hair, Blonde Hair}. Conditioning concepts in CelebA are quantized with less information about the degree of smiling or hair color, which results in harder convergence for novel concept generation. These attributes are treated as binary concept tuples similar to the Shapes2D setting, enabling consistent evaluation across synthetic and real-world domains. We curate 10,000 images per concept class, yielding a total of 40,000 training examples. This setup allows us to examine how well models trained under structured compositional constraints in CelebA can generalize to novel combinations of facial attributes.

### B.4 CLEVRER-Kubric

The CLEVRER-Kubric dataset is a reimplementation of CLEVRER Yi et al. (2020) from scratch using Kubric Greff et al. (2022). This dataset is generated by simulating physics-based scenes using Blender. The scenes are configured with parameters similar to CLEVRER, including a resolution of $480 \times 320$ pixesl, a frame rate of 12 fps, and a simulation step rate of 240Hz. Each scene includes a static floor with a gray material and a directional light source. A perspective camera is positioned to view the scene. Stationary objects, numbering between 5 and 7, are randomly chosen from cubes or spheres, and are placed within a pre-defined spawn region. The material properties for individual objects are controlled based on the composition being tested (that is, all instances of a specific color or shape can be held out to ensure that the model does not see a specific combination during training).

The dynamic aspect involves 1 to 3 projectiles, also from the pre-defined shapes, with similar controlled material and size properties. The projectiles are launched from random points around the perimeter with a velocity between 10 and 15 units, slight randomness to the trajectory, and zero angular velocity. There is increased angular damping to maintain consistency with the original CLEVRER videos. Each composition has 250 videos, each with 28 frames, yielding 7000 frames per composition.

As with Shapes2D, we control for shape={cube, sphere}, size={large, small}, color={red, green}. For every composition, we put the desired composition as one (or more) of the static objects to force the model to generate that (and not learn a shortcut by simply generating other parts of the video). We increase the probability of having the dynamic moving object also follow the same properties as the composition to ensure that our methods also learn to model temporal consistency for compositions that are out of the training distribution.

## C Experiments

### C.1 Tokenizer

Similar to Section 4.1 of the main text, we also trained MAR with three different types of tokenization schemes to prove that the choice of tokenizer does not alter the compositional capabilities of a model. We show linear probe results in Figure 7. By showing a similar trend, we complement the experiments on DiT shown in the main text, and also show that the choice of downstream generative architecture is not a confounding factor when comparing different tokenizers. To further reinforce the generality of our experiments and demonstrate that the same trend extends to videos, we also present results on CLEVRER-Kubric using a DiT model trained under three different tokenization schemes (Figure 8).

**Mapping discrete tokens to continuous vectors for downstream models** When employing VQ-VAE tokenizer with models like DiT and MAR, the process involves an intermediate step to bridge the discrete nature of VQ-VAE tokens with the continuous vector processing expected by these architectures. *For our experiments in the main text, we use the continuous embedding vectors before the quantization step in the tokenizer.* However, we add a learnable embedding layer in our pipeline for our experiments on CLEVRER-Kubric. The VQ-VAE first encodes an input data sample into a

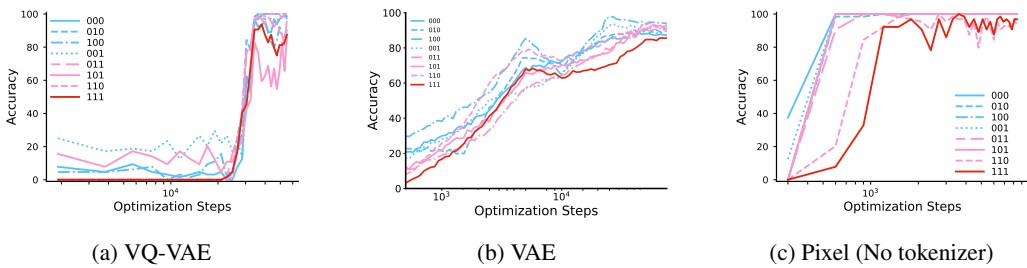

(a) VQ-VAE        (b) VAE        (c) Pixel (No tokenizer)

Figure 7: MAR exhibits generalization on both level-1 and level-2 compositions in Shapes2D regardless of whether (a) discrete, (b) continuous, or (c) no tokenizer is used. The blue curves show performance on training data, pink curves depict level-1 compositions, and red curve denotes level-2 compositions.

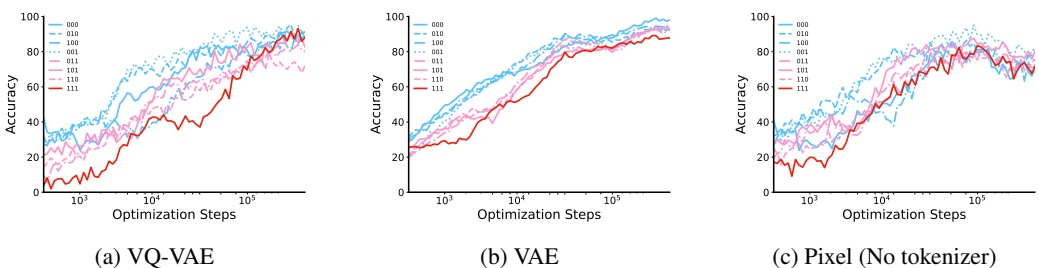

(a) VQ-VAE        (b) VAE        (c) Pixel (No tokenizer)

Figure 8: DiT exhibits generalization on both level-1 and level-2 compositions in CLEVRER-Kubric regardless of whether (a) discrete, (b) continuous, or (c) no tokenizer is used. The blue curves show performance on training data, pink curves depict level-1 compositions, and red curve denotes level-2 compositions.

spatial grid of discrete token IDs, where each ID corresponds to a learned codebook vector. For a downstream model like a transformer, these discrete token IDs are then passed through a learnable embedding layer. This layer maps each token ID to a unique, dense, continuous vector representation. It is this grid of continuous vectors (token embeddings) that then serves as the actual input to the downstream generative model. This embedding step effectively translates the discrete symbolic representation from the VQ-VAE into a continuous vector space where the model can perform its computations to achieve generation and composition.

## C.2 TRAINING OBJECTIVE

Complementing results from Shapes2D Okawa et al. (2024) and CelebA Liu et al. (2018) in the main text, we show that our findings also hold true for more complex modalities like video. Figure 9 contrasts the performance of a discrete objective (MaskGIT) against a continuous objective (DiT) on the CLEVRER-Kubric dataset. We see the same trend of MaskGIT showing significantly worse performance, especially on level-2 compositions. Following the structure in the main text, we also show results on MAR (Figure 9c) to disentangle the effects of the architectural design from the properties of the latent space and further reinforce our results that objectives modeling underlying continuous factors achieve better compositionality.

## C.3 CONDITIONING INFORMATION

Section 4.3 examined the effect of different conditioning mechanisms on a DiT trained on Shapes2D. Figure 4d illustrates a memorization failure mode that arises when concept information is only partially provided during training. In such cases, the model tends to generate realistic images that are misaligned with the intended conditioning concept but resemble the closest training instance, e.g., a blue big triangle is rendered as a red big triangle. Figure 10 shows the effect of different conditioning mechanisms for a DiT trained on CLEVRER-Kubric. Given the difference in modalities, we slightly alter our experimental setup to make use of the temporal component in videos that provide

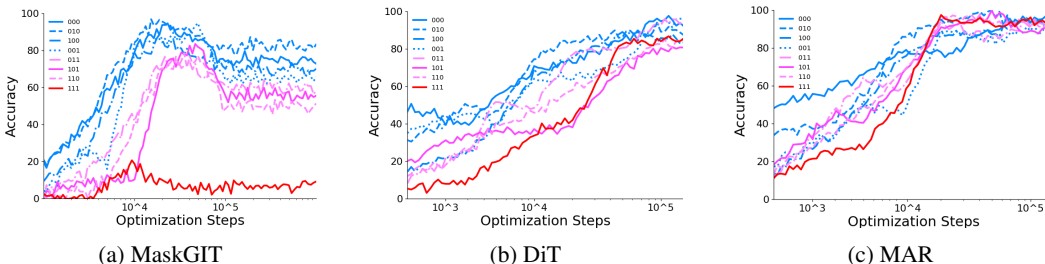

| (a) MaskGIT | (b) DiT | (c) MAR |

Figure 9: Comparison of linear probe accuracy for CLEVRER-Kubric across (a) MaskGIT, (b) DiT, and (c) MAR. Models leveraging a continuous latent space (DiT, and MAR) show better level-2 compositions than MaskGIT. The blue curves show performance on training data, pink curves depict level-1 compositions, and red curve denotes level-2 compositions.

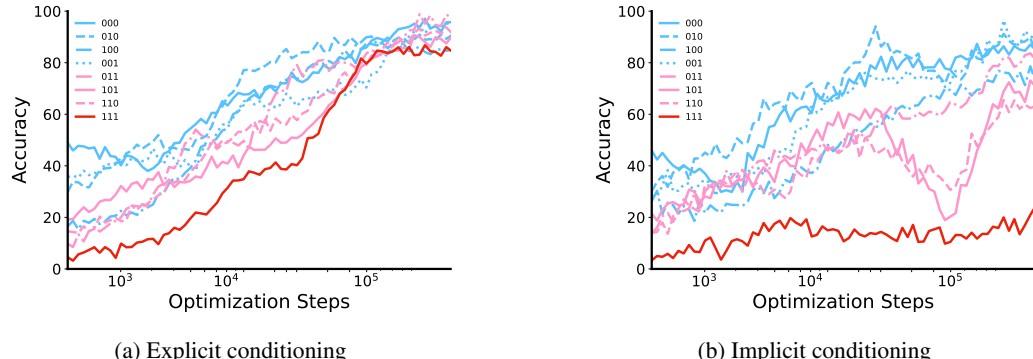

| (a) Explicit conditioning | (b) Implicit conditioning |

Figure 10: Comparison of linear probe accuracy for DiT trained on CLEVRER-Kubric with (a) Explicit conditioning and (b) Implicit conditioning mechanisms. Training with explicit label information allows for better level-2 compositions. The blue curves show performance on training data, pink curves depict level-1 compositions, and red curve denotes level-2 compositions.

a pre-defined context in the form of past frames in which concept information is presented implicitly. For our experiments on CLEVRER-Kubric, we evaluate whether compositional abilities depend on concept information provided explicitly as labels (Figure 10a) or implicitly through context frames (Figure 10b). The results show that explicit signals are crucial for compositional generalization, highlighting the models' limitations in abstracting concept information from implicit observations alone. This finding parallels our Shapes2D results, suggesting that when concept information is partially or not explicitly provided, the model must extract it from raw observations in pixel space, a challenge usually present in representation learning that often limits compositional generalization.

# D  ENHANCING COMPOSITIONAL LEARNING WITH JOINT EMBEDDING PREDICTIVE ARCHITECTURES

## D.1  AUGMENTING THE MASKGIT TRAINING OBJECTIVE WITH JEPA

The core idea of JEPA is to predict the *representation* of masked portions of the input (target) blocks from the *representation* of visible portions (context blocks) in an abstract embedding space and to learn both the representational embedding *and* the predictor end-to-end. By positioning the target not as a noisy, high-dimensional input, but in a lower-dimensional, cleaned-up embedding, it allows the model to capture relevant factors in the target content. This encourages the encoder to learn representations that capture high-level, predictable semantic information while potentially abstracting away low-level, instance-specific details that may hinder generalization. We formulate our JEPA-based training objective described below.

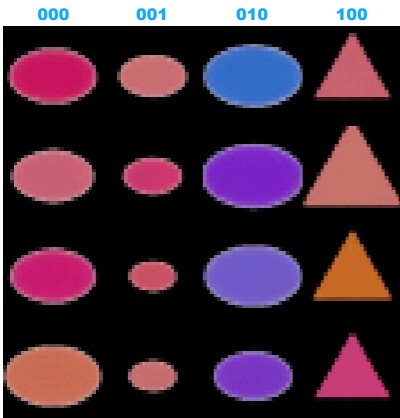 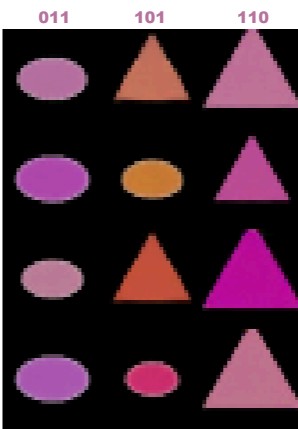 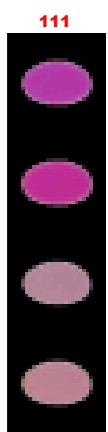

Figure 11: Qualitative Results of the concept dropout with DIT on Shapes2D. The visualization demonstrates that when contextual information about all concepts is incomplete, the model struggles to generalize compositionally. A typical failure mode is the substitution of novel concept combinations with the closest seen ones—for instance, the 110 configuration, which should correspond to a novel blue big triangle, is instead rendered as a red big triangle, likely due to its proximity in the training distribution. Similar behavior is happening to the 101, which should correspond to a novel red small triangle, sometimes result in a red small circle or red big triangle.

An input data sample $x$ is first encoded into a sequence of discrete visual tokens $E_{VQ}(x) \in \{1, ..., K\}^N$ where $E_{VQ}$ is the encoder trained with a VQ-VAE objective and $N$ is the sequence length. The standard MaskGIT objective ($\mathcal{L}_{MG}$) focuses on reconstructing masked input tokens. We employ block-masking Bardes et al. (2024) for images and causal block masking for videos Bardes et al. (2024) to sample a mask $\mathcal{M} \subset \{1, ..., N\}$. The MaskGIT Transformer $T$ predicts the probability distribution $p(z_i|z_{\setminus M})$ for each masked token $z_i$ ($i \in M$) based on unmasked token $z_{\setminus M}$. The loss is the negative log-likelihood of the true tokens at the masked positions:

$$\mathcal{L}_{MG} = -\mathbb{E}_{x,M}\left[\sum_{i \in M} \log p(z_i|z_{\setminus M})\right] \tag{1}$$

Let $H^{(l)}$ represent the sequence of continuous hidden state vectors output by the $l$-th layer of the MaskGIT Transformer $T$. We introduce a JEPA-style objective applied to the hidden states of specific intermediate layers $l \in L_{JEPA}$. This acts as a representation alignment objective within the network.

For each selected layer $l \in L_{JEPA}$, we apply a separate JEPA masking strategy $M_{JEPA}$. This strategy defines a set of context indices $C \subset \{1, ..., N\}$ and target indices $T \subset \{1, ..., N\}$, typically corresponding to spatial blocks. We extract the hidden states from layer $l$: $H^{(l)} = \{h_1^l, ..., h_N^l\}$. The extracted *context* hidden states $H_C^{(l)} = \{h_j^{(l)}|j \in C\}$ is used as input to a layer-specific predictor network $P_{JEPA}^{(l)}$. In practice, $P_{JEPA}^{(l)}$ is simply parametrized using a multilayer perceptron (MLP). The predictor network aims to predict the hidden states of the target tokens $\hat{H}_T^{(l)} = P_{JEPA}^{(l)}(H_C^{(l)})$.

The JEPA loss for layer $l$ measures the discrepancy between the predicted target representation (for target index $k \in T$) $\hat{h}_k^{(l)}$ and the actual target representations $h_k^{(l)}$ (computed by the Transformer $T$ itself) using Mean Squared Error (MSE) as a distance metric $d$

$$\mathcal{L}_{JEPA}^{(l)} = \mathbb{E}_{x,M_{JEPA}}\left[\frac{1}{|T_{idx}|}\sum_{k \in T_{idx}} d\left(\hat{h}_k^{(l)}, sg(h_k^{(l)})\right)\right] \tag{2}$$

where $T_{idx}$ denotes the set of target indices and $sg(\circ)$ denotes the stop-gradient operation. This ensures that the encoder is updated to produce context representations $H_C^{(l)}$ that are predictive of the target representations $h_k^{(l)}$, rather than trivially learning to reconstruct $h_k^{(l)}$ through the target pathway.

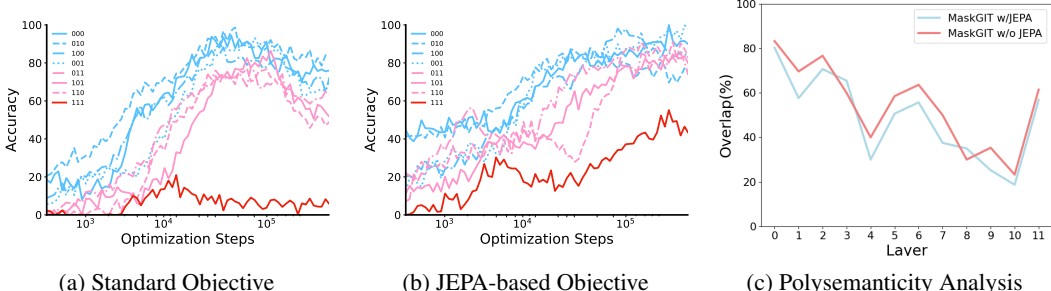

(a) Standard Objective  (b) JEPA-based Objective  (c) Polysemanticity Analysis

Figure 12: Comparison of linear probe accuracy for MaskGIT with Standard (a) and JEPA-based (b) training objectives for CLEVRER-Kubric. The JEPA-based training objective clearly enhances compositional abilities, and lowers polysemanticity (c) even though it cannot fully compensate the problems introduced by the discrete space. The blue curves show performance on training data, pink curves depict level-1 compositions, and red curve denotes level-2 compositions.

The final JEPA loss can be the same as Equation 2 if applied only to a single layer, or a summation of losses applied to all layers. In the latter case:

$$\mathcal{L}_{JEPA} = \sum_{l \in L_{JEPA}} \mathcal{L}_{JEPA}^{(l)} \tag{3}$$

The final training objective is a weighted sum of the standard MaskGIT reconstruction loss (Equation 1 and the JEPA representation alignment loss (Equation 3):

$$\mathcal{L}_{Total} = \mathcal{L}_{MG} + \lambda \mathcal{L}_{JEPA} \tag{4}$$

where $\lambda$ is a hyperparameter balancing the contribution of the two objectives.

### D.2 RESULTS ON CLEVRER-KUBRIC

To reinforce our findings from the main text, we also extend our JEPA-based objective for video generation using the CLEVRER-Kubric dataset. We see that our results for a similar trend as reported for Shapes2D in the main text. Figure 12a shows that the standard MaskGIT objective is insufficient for reliable compositional generalization. However, augmenting it with our JEPA-based objective yield significant improvement in compositional abilities (Figure 12b). We also conduct a polysemanticity analysis (Figure 12c), observing the same trend as Shapes2D, with the JEPA-augmented objective exhibiting lower polysemanticity in attention heads- which points to the model learning factor-specific representations.

### D.3 ABLATION STUDIES

**Analysis on JEPA-loss on different layers**  As seen in the previous sections, we apply the JEPA loss to different layers $l$ within the transformer block. We show an empirical ablation below contrasting the performance of applying the JEPA loss to different layers for a MaskGIT model trained on CLEVRER-Kubric Table 2 shows the maximum probe accuracy of the `111` composition with the JEPA losses at different layers or combinations of layers. Following previous works Yu et al. (2025), we also applied predictive coding in the representations of mid-level layers. We see slight differences after varying combinations but we phrase the problem as parameter tuning and empirically select the best combination for optimal trade-off between accuracy and performance.

**Effect of $\lambda$**  We also provide an empirical ablation on varying the weighting factor $\lambda$ as seen in Equation 3.

Table 2: Maximum linear probe accuracy after applying the JEPA loss on different layers and combinations of layers for a MaskGIT model trained on CLEVRER-Kubric

| Layers | Accuracy |
|--------|----------|
| {6} | 27.61 |
| {8} | 26.35 |
| {6,8} | 38.16 |
| {7,9} | 36.62 |
| {6,8,10} | 53.52 |
| {7,9,11} | **56.27** |
| {7,8,9,11} | 47.25 |

Based on the graph, we select 0.6 as the weighting factor for our experiments.

### D.4 POLYSEMANTICITY IN ATTENTION HEADS

We give a deeper explanation into how we perform the polysemanticity analysis mentioned in the main text. We quantify polysemanticity in attention heads by assessing their causal impact on the model's ability to distinguish between different visual features like color or shape. An attention head is identified as polysemantic if it significantly influences the processing of multiple distinct features. The core of this ap-

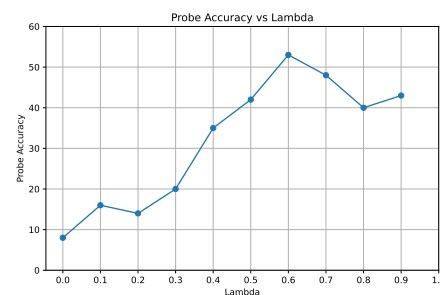

Figure 13: Effect of weighting factor on probe accuracy

proach involves a causal intervention- specifically, ablating individual attention heads and measuring the resultant change in the model's representation for contrasting pairs of data samples.

First, we establish a baseline for how the original, unablated model represents and distinguishes the features of interest. We do so by passing curated pairs of images or videos, with the samples in each pair differing along a single, targeted visual feature dimension. For instance, to assess color processing, we use pairs of red square and blue square. For each sample $I$, its global representation is extracted from the model, typically by using an additional token that aggregates all features of the sample (similar to the [CLS] token). These embeddings are L2 normalized before further use.

$$[CLS]_{norm}(I) = \frac{[CLS](I)}{||[CLS](I)||_2} \tag{5}$$

The model's ability to distinguish a feature $f$ is quantified by the cosine similarity between these embeddings for pair $(I_a, I_b)$ contrasting in that specific feature. Given that the tokens are already L2-normalized, their cosine similarity is simply their dot product

$$sim(I_a, I_b) = [CLS]_{norm}(I_a) \cdot [CLS]_{norm}(I_b) \tag{6}$$

This gives us a baseline similarity score $sim_{baseline,f}$ for each feature under consideration.

Next, we perform targeted ablation of a specific attention head $h$ within a given layer $l$. This intervention involves creating a copy of the model and modifying the forward pass of its multi-head self-attention (MHSA) module. Specifically, the $QKV$ vectors corresponding to $h$ are set to 0 immediately before the attention scores are computed. This allows us to observe the model's behavior in the absence of the specific attention head $h$.

Using the ablated model, $M_{ablated(l,h)}$, we recalculate the L2-normalized [CLS] token embeddings and subsequently the feature similarity score $sim_{ablated,f}$ for the same pairs. The causal impact

$E_f(l, h)$ of ablating head $(l, h)$ on the processing of feature $f$ is defined as the change in this similarity score:

$$E_f(l, h) = sim_{ablated,f} - sim_{baseline,f} \qquad (7)$$

An attention head $(l, h)$ is considered to significantly impact feature $f$ if the absolute magnitude of its causal effect $|E_f(l, h)|$, exceeds a predefined threshold $\tau$. For our experiments, we consider $\tau = 0.005$. The head is then defined polysemantic if it meets this significance criterion for multiple distinct features.

### D.5 MECHANISTIC SIMILARITY

We provide a more comprehensive explanation of our method to compare the circuit overlap for different concepts across different models. Similar to Kempf et al. (2025), we first identify important neurons within the MLP blocks by perturbing their activations. Specifically, we zero out the output of the fully connected layer for a neuron and measure the impact on the model's classification score for a target class. The impact is defined as

$$\text{Impact}(a_{l,j}) = |S_C(I) - S'_C(I|a_{l,j} = 0)| \qquad (8)$$

where $S_C(I)$ is the baseline score for class $C$ given data sample $I$ and $S'_C(I|a_{l,j} = 0)$ is the score when neuron $j$'s activation in block $l$ is zeroed out ($a_{l,j} = 0$). The top $10\%$ neurons causing the largest score change are selected.

Next, we identify important connections between these MLP neurons across consecutive transformer blocks (from block $k$ to $k + 1$). We attribute the activation of a target important neuron in the fully connected layer of block $k + 1$ to the activations of neurons in block $k$. A connection is deemed important if the source neuron has high attribution and was also identified as an important neuron.

Finally, these important neurons and connections form a class specific circuit-graph (visualized in Figure 14). We then compare circuits from different classes using Jaccard similarity for measuring neuron overlap.

## E EVALUATION

### E.1 SHAPES3D

| Model | (0,0,0) | (0,0,1) | (0,1,0) | (1,0,0) | (1,0,1) | (1,1,0) | (0,1,1) | (1,1,1) |
|---|---|---|---|---|---|---|---|---|
| DiT | 100% | 100% | 100% | 100% | 100% | 100% | 100% | 100% |
| MaskGIT | 94.7% | 97% | 94.7% | 94% | 79.49% | 62.6% | 58.85% | 30% |

Table 3: Performance comparison of DiT and MG across different compositions.

We follow the Shapes2D evaluation protocol and measure compositional generalization using probe accuracy. The probe is a lightweight ResNet-style classifier trained to predict object properties from generated images. It consists of three convolutional blocks with residual connections, followed by global average pooling. The output is passed to three separate linear heads that predict the object's hue, shape, and scale.

As shown in Table 3, our key finding still holds: DiT (continuous) generalizes effectively to novel compositions, while MaskGIT (discrete) struggles. Notably, MaskGIT exhibits failure cases such as non-monotonic color attribution to objects and regression to nearest-neighbour training examples.

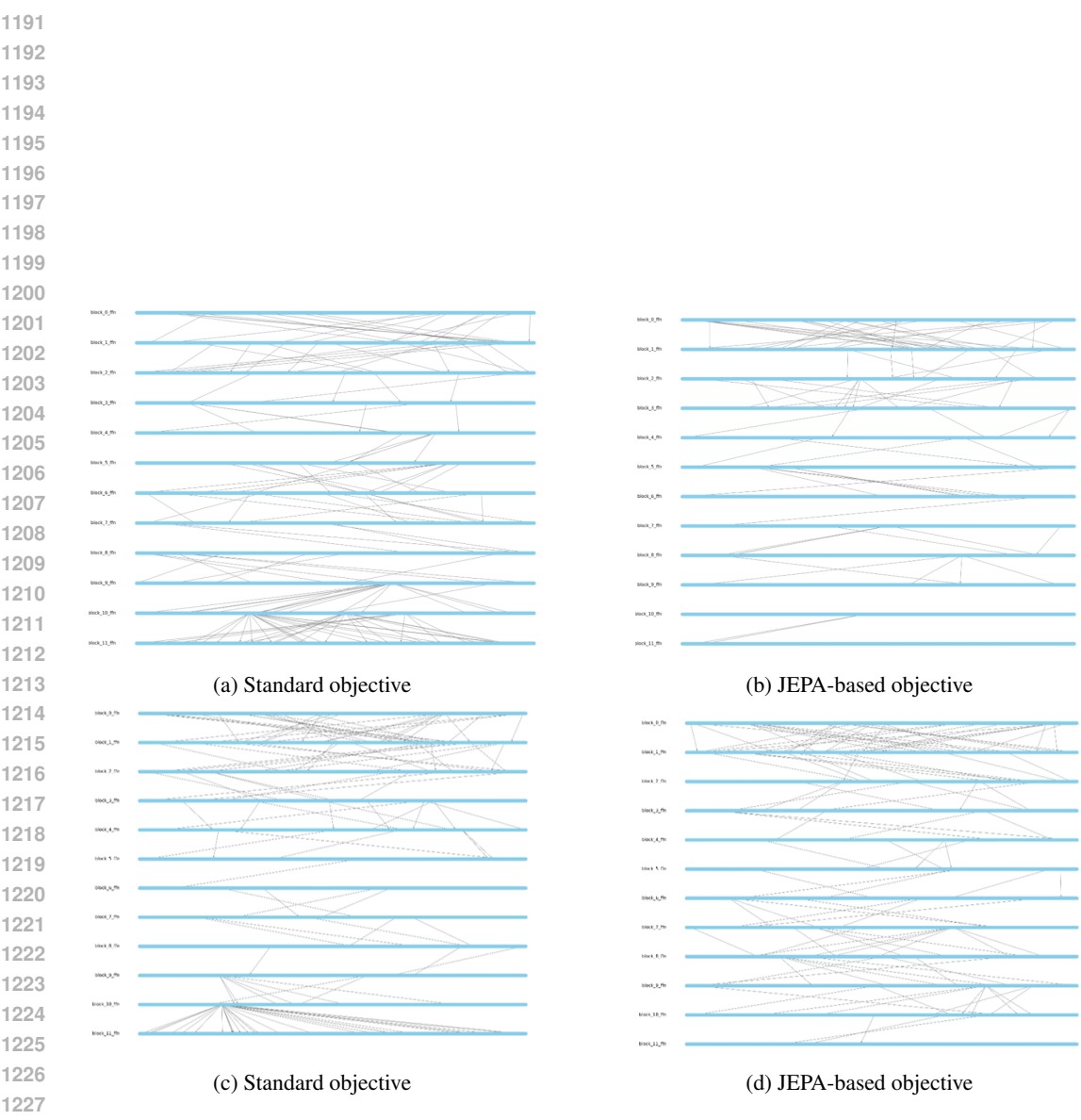

(a) Standard objective

(b) JEPA-based objective

(c) Standard objective

(d) JEPA-based objective

Figure 14: Visualization of MaskGIT circuits for a `large red cube` and `small blue sphere` trained with the standard training objective versus the JEPA-based training objective.

| Factor | Possible Values | Super Group 0 | Super Group 1 |
|--------|-----------------|---------------|---------------|
| Color | red, orange, yellow, lime green, green, cyan, blue, indigo, purple, pink | red, orange, yellow, lime green, green, cyan, blue | indigo, purple, pink |
| Shape | cube, cylinder, sphere | cube, cylinder | sphere |
| Size | 0–7 (ordinal) | 0, 1, 2, 3, 4, 5 | 6, 7 |

Table 5: Factorization of the data into *Super Groups*. This partitioning is used to evaluate compositional generalization across disjoint super groups.

### E.2 CELEBA

Since there are no established metrics that effectively measure compositional generalization in visual generative models, we introduce a retrieval-based evaluation metric that assess the compositionality. The core idea is to test whether generated samples from an unseen composition (e.g., `111`, two factors away from the trainset) are closest in feature space to real images of the same composition. We compute DINOv2 Oquab et al. (2024) distances be-

Table 4: Compositional Retrieval Accuracy (CRA, %) 100 generated samples.

| Model | 1-NN | 5-NN |
|-------|------|------|
| DiT | 27 | **36** |
| MaskGIT | 21 | 16 |
| MaskGIT + JEPA | **31** | 27 |

tween each generated image and all real images for the eight compositions. Using Nearest Neighbor retrieval accuracy, we measure how often a generated image is closest to real images from its target composition. We evaluate DiT, MaskGIT, and JEPA-enhanced MaskGIT on samples from the unseen composition `111`. Models leveraging continuous distributions show a clear advantage in compositional generalization (Table 4).

## F    RESULTS

### F.1    SHAPES2D

Figure 15 gives a comprehensive overview of qualitative results on Shape2D.

### F.2    SHAPES3D

Figures 16 and 17 gives a comprehensive overview of qualitative results on Shapes3D. For easier indication of compositional novelty, we use padding colors based on the sum of supergroup assignments across the three factors (color, shape, size), where each factor contributes 0 or 1 depending on its supergroup. Dark blue indicates a sum of 0 (all factors from Supergroup 0), light blue indicates a sum of 1, pale orange corresponds to a sum of 2, and dark red represents a sum of 3 (all factors from Supergroup 1, i.e., the most novel compositions).

### F.3    CELEBA

Figure 18 gives a comprehensive overview of qualitative results on CelebA.

### F.4    CLEVRER-KUBRIC

Qualitative results on CLEVRER-Kubric are presented as a separate website in the included HTML file. We show results for `111 - {large red cube}`, and can see that MaskGIT trained with the standard objective generates colors well but struggles to assign them to specific shapes. In contrast, MaskGIT trained with a JEPA-based objective and models like DiT which leverage a continuous latent space show better performance, as reflected by the probe accuracies (Figure 9).

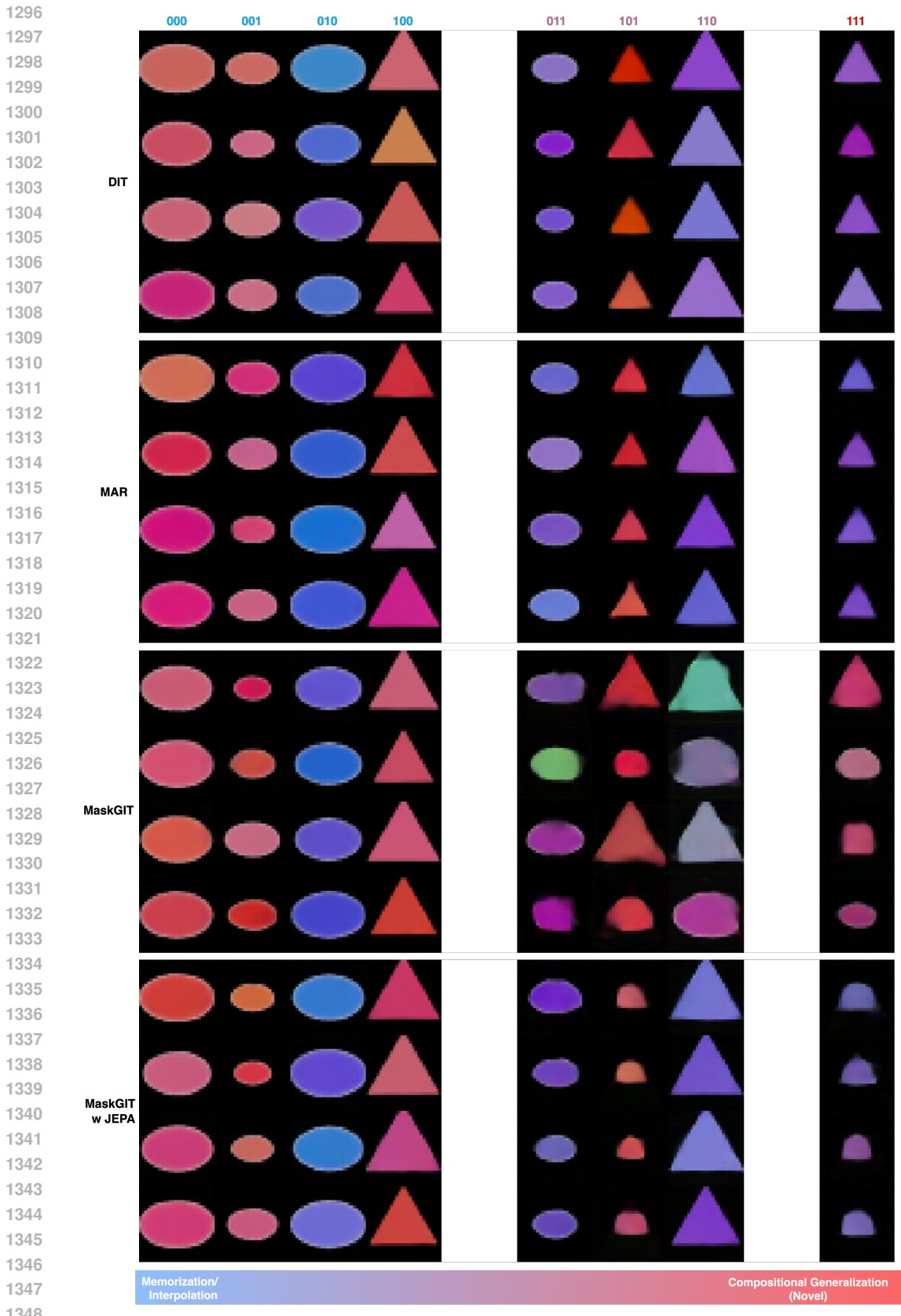

Figure 15: Qualitative Results on Shapes2D

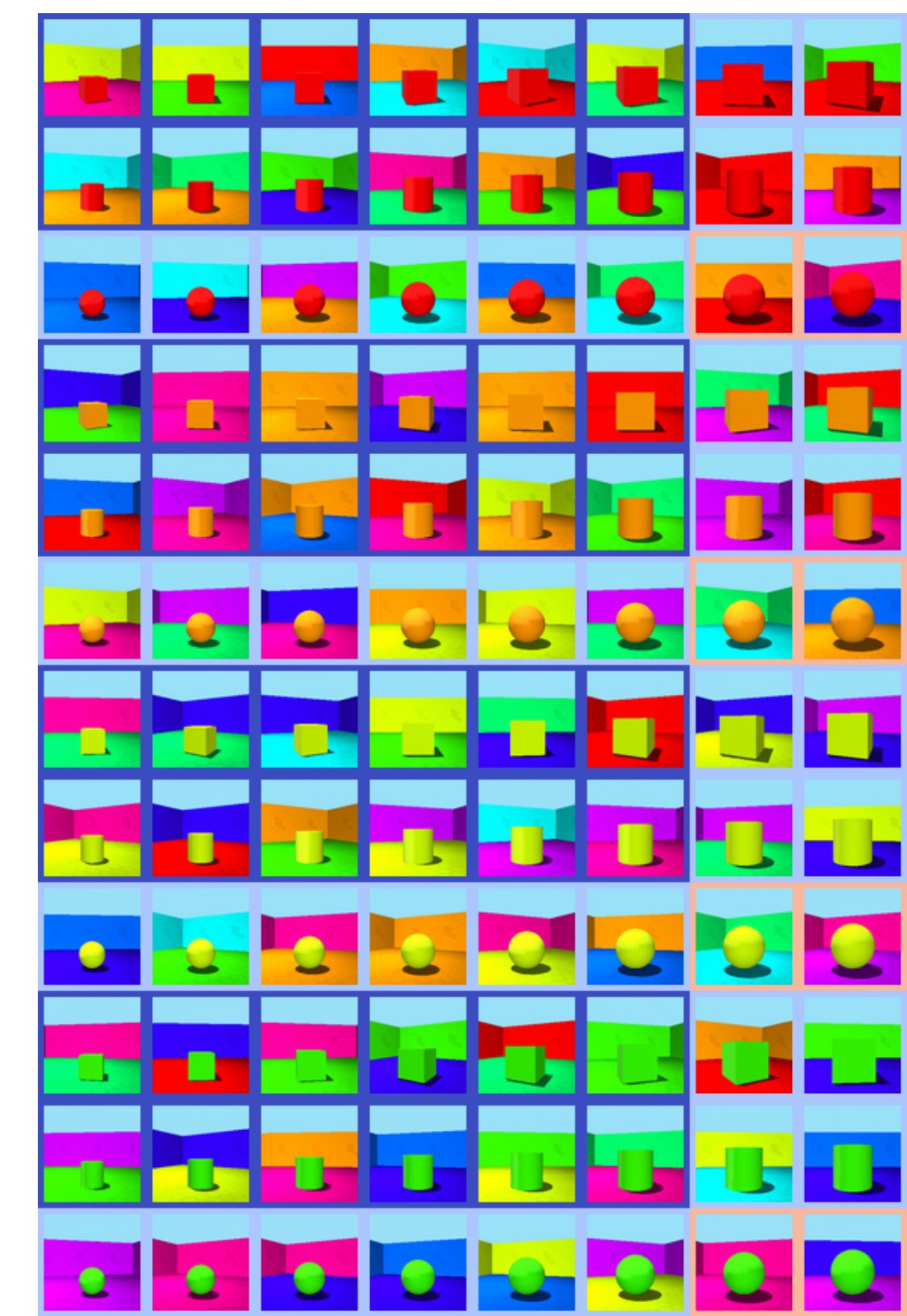

Figure 16: Shapes3D results of DiT. Dark blue: all factors from Supergroup 0; light blue: one factor from Supergroup 1; pale orange: level-1 compositions; and dark red: level-2 novel compositions.

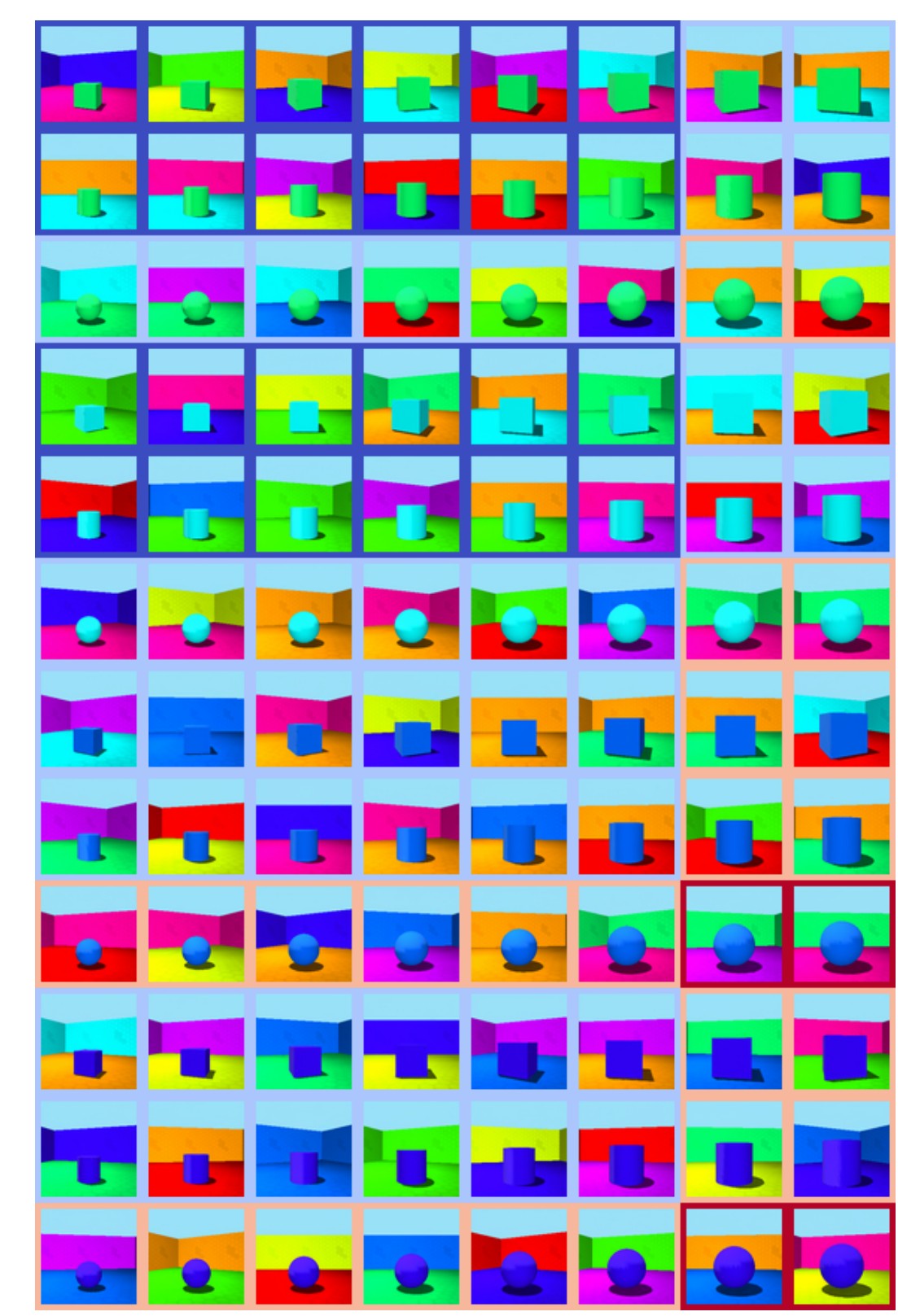

Figure 16: Shapes3D results of DiT. Dark blue: all factors from Supergroup 0; light blue: one factor from Supergroup 1; pale orange: level-1 compositions; and dark red: level-2 novel compositions.

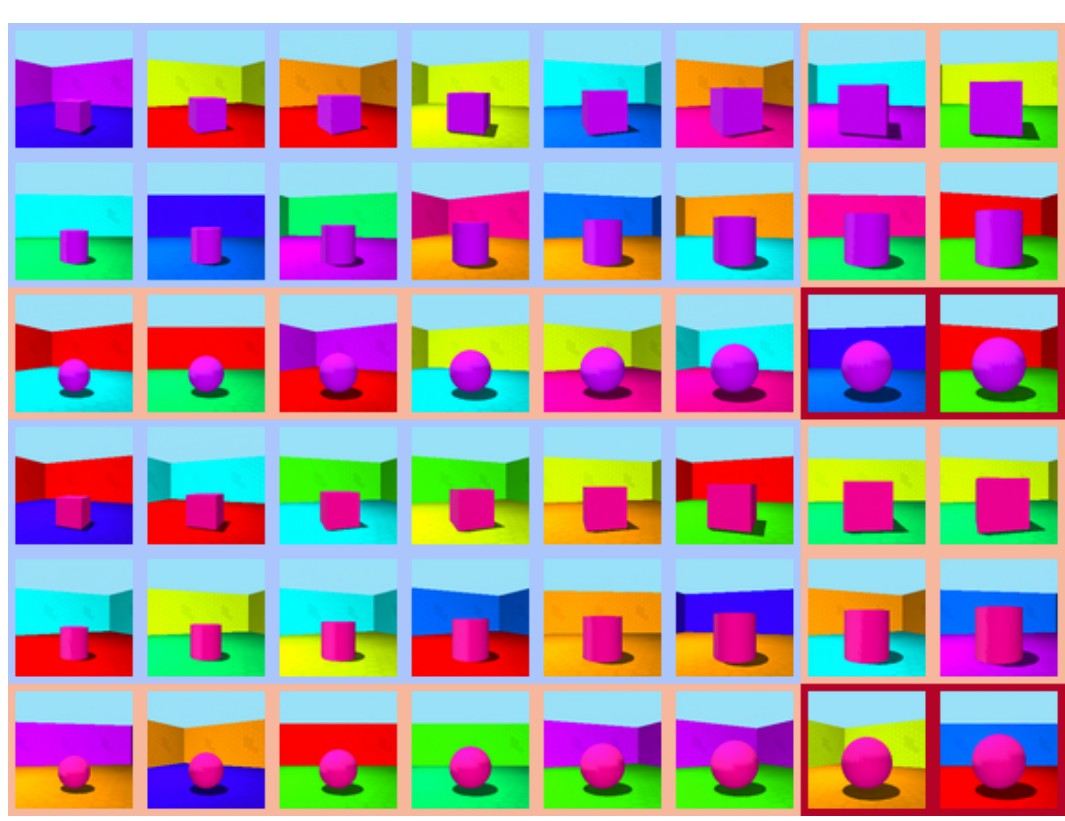

Figure 16: Shapes3D results of DiT. Dark blue: all factors from Supergroup 0; light blue: one factor from Supergroup 1; pale orange: level-1 compositions; and dark red: level-2 novel compositions.

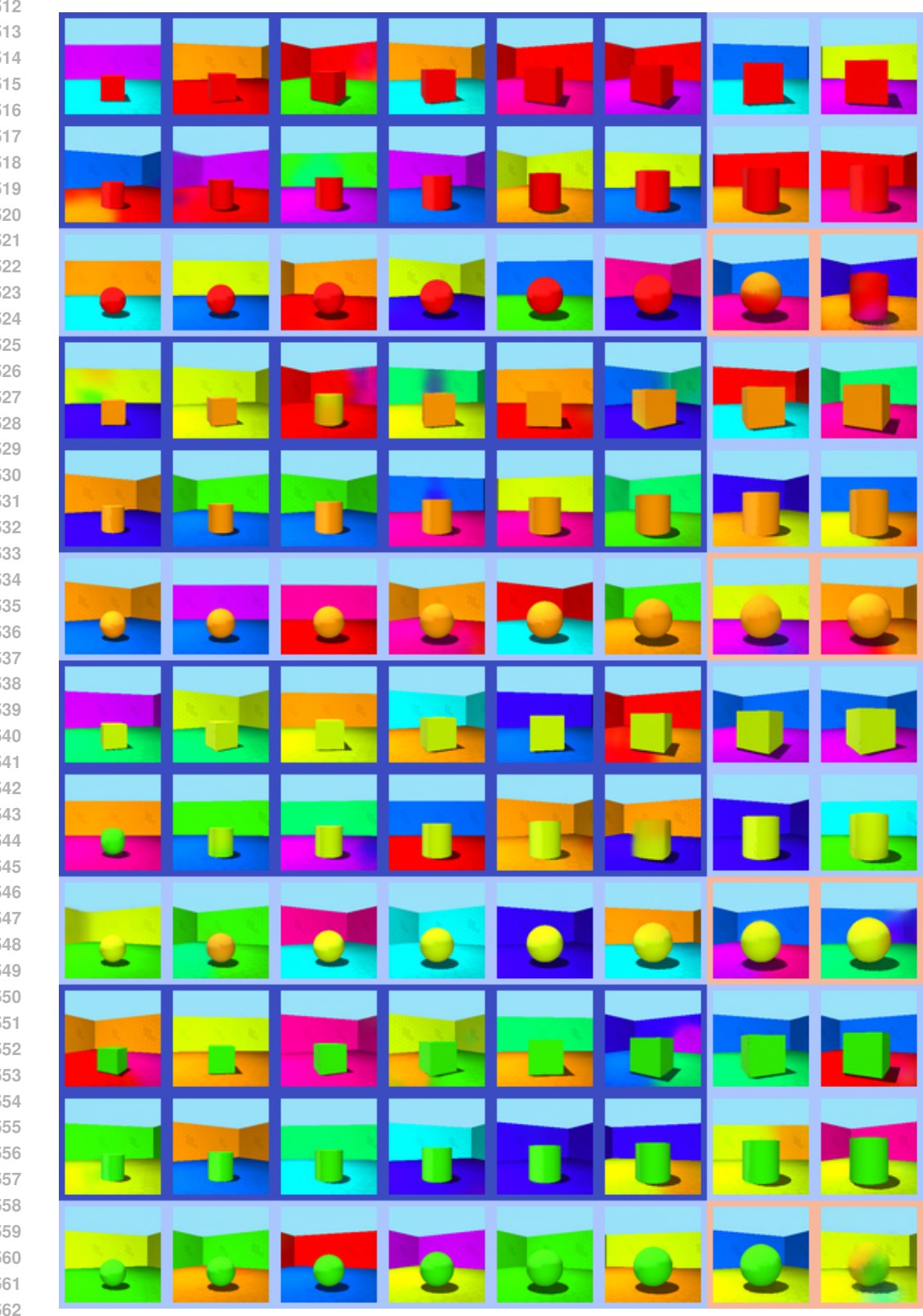

Figure 17: Shapes3D results of MaskGIT. Dark blue: all factors from Supergroup 0; light blue: one factor from Supergroup 1; pale orange: level-1 compositions; and dark red: level-2 novel compositions.

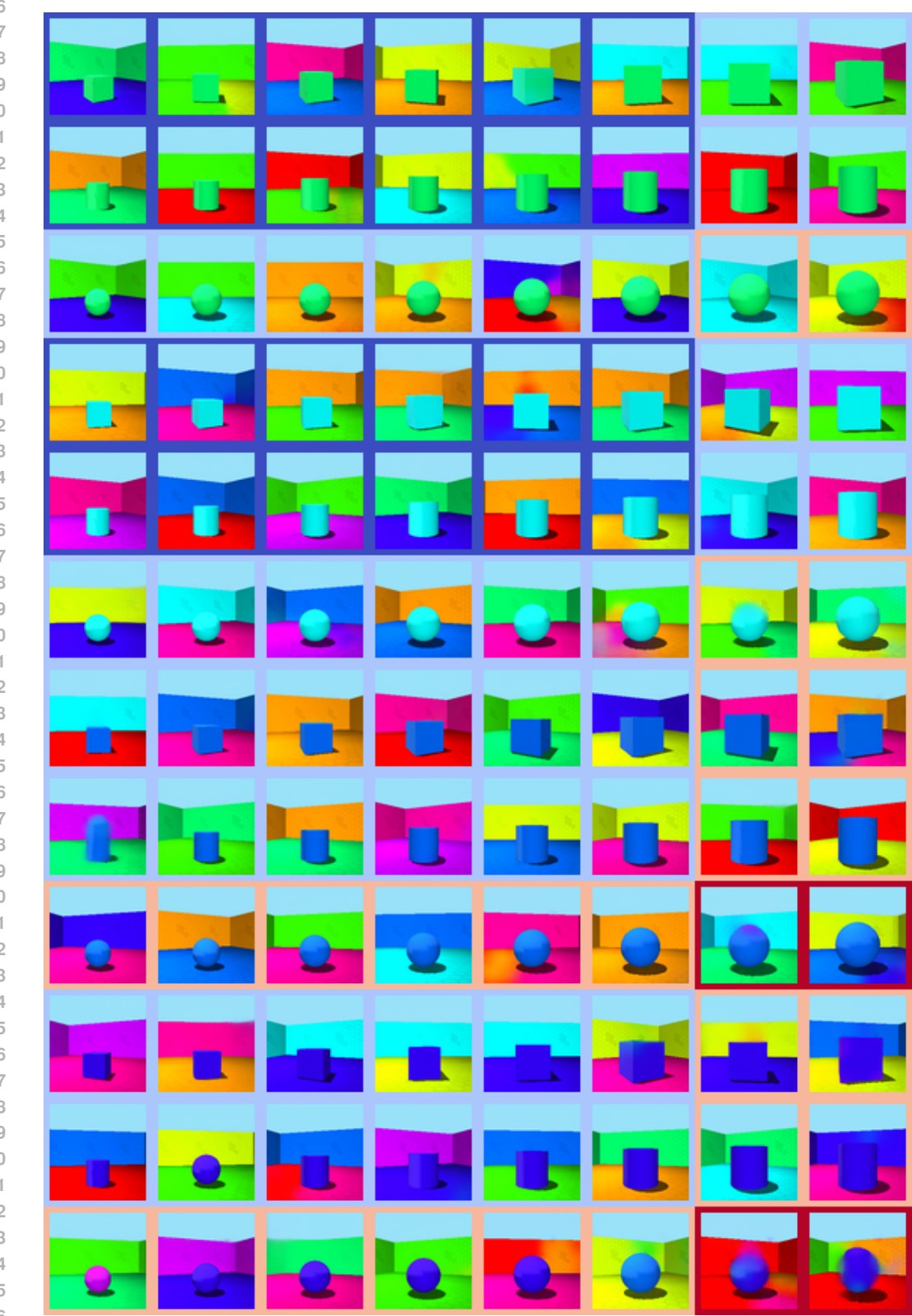

Figure 17: Shapes3D results of MaskGIT. Dark blue: all factors from Supergroup 0; light blue: one factor from Supergroup 1; pale orange: level-1 compositions; and dark red: level-2 novel compositions.

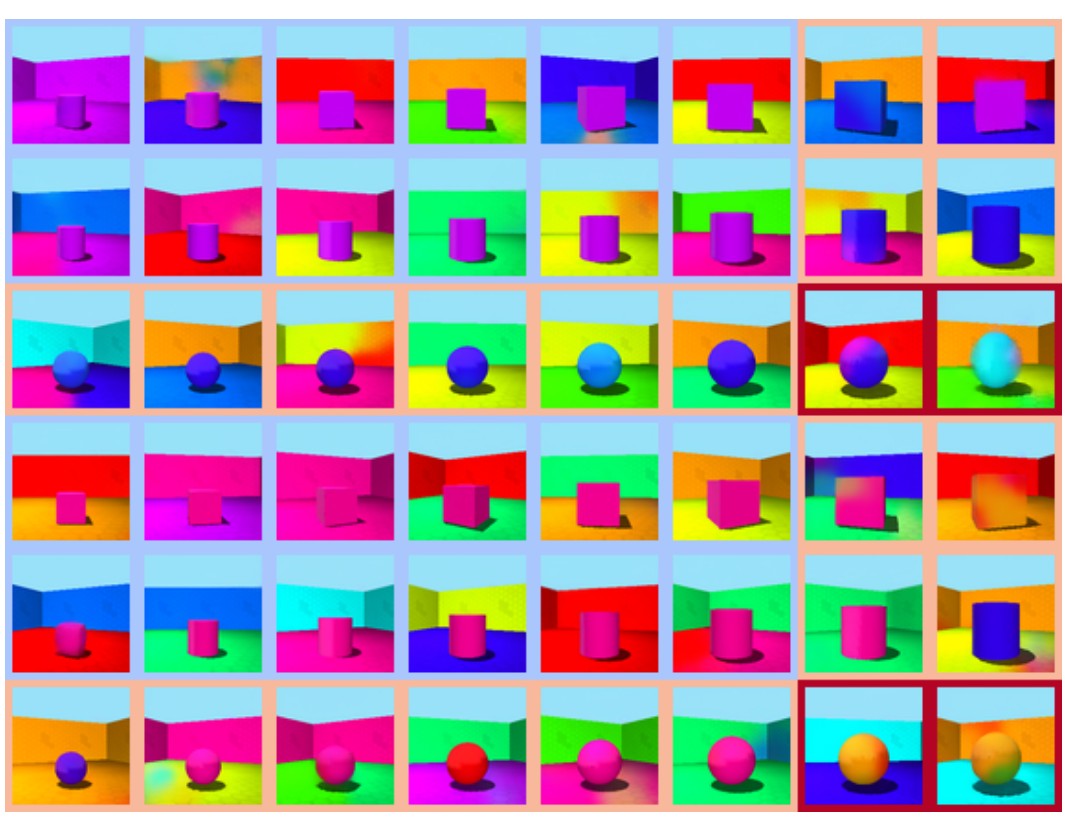

Figure 17: Shapes3D results of MaskGIT. Dark blue: all factors from Supergroup 0; light blue: one factor from Supergroup 1; pale orange: level-1 compositions; and dark red: level-2 novel compositions.

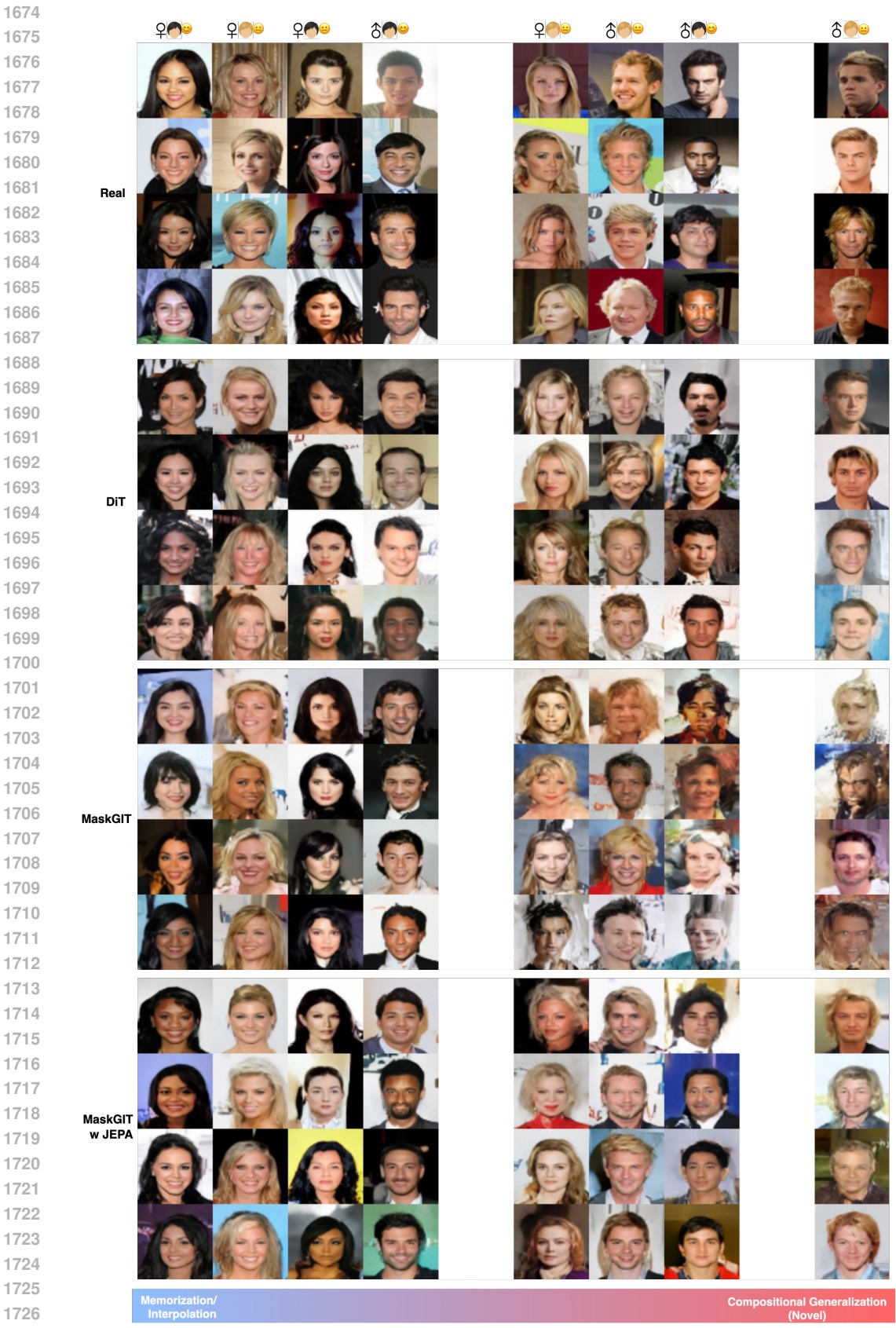

Figure 18: Qualitative Results on CelebA

Table 6: Nearest-neighbor compositional retrieval metric (CRA) comparing generated videos to real videos. Factors are time of day: Day $\odot$, Night $\bigcirc$; directions: Left $\leftarrow$, Straight $\uparrow$, Right $\rightarrow$. Rows show a generated novel split evaluated against *real train* and *real novel* classes. The Hit@1 (correct target novel class) is bolded; **black bold** indicates better results.

| | | Real train | | | | Real novel | |
|---|---|---|---|---|---|---|---|
| Gen. novel | Model | $\odot\leftarrow$ | $\odot\uparrow$ | $\bigcirc\rightarrow$ | $\bigcirc\uparrow$ | $\odot\rightarrow$ | $\bigcirc\leftarrow$ |
| $\odot\rightarrow$ | DiT | 0.190 | 0.190 | 0.060 | 0.060 | **0.470** | 0.030 |
| | MaskGIT | 0.380 | 0.260 | 0.110 | 0.050 | 0.180 | 0.020 |
| $\bigcirc\leftarrow$ | DiT | 0.130 | 0.020 | 0.340 | 0.040 | 0.040 | **0.430** |
| | MaskGIT | 0.150 | 0.030 | 0.560 | 0.100 | 0.020 | 0.140 |

# G  WORLDS MODELS: COVLA

We use CoVLA (Arai et al., 2025), a real-world driving video dataset collected in Tokyo with synchronized front-facing camera and CAN/GNSS/IMU signals. We do the compositional splits on two factors: time of day (day $\odot$, night $\bigcirc$) and turn direction (left $\leftarrow$, straight $\uparrow$, right $\rightarrow$). The held-out novel compositions are $\odot\rightarrow$ and $\bigcirc\leftarrow$; all remaining combinations constitute the *real-train* set.

**Labeling and segmentation.**  We assign the night label using `in_tunnel` flag or explicit night time at the exact timestamp for the frame, label *turns* when speed $\geq 1.8\,\mathrm{km/h}$ and steering angle $\geq 10°$ (left/right), and segment videos so each clip contains exactly one scenario (no factor changes within a clip).

**Models and training.**  We instantiate ORBIS (Mousakhan et al., 2025) in two matched variants: Orbis-MaskGIT (categorical objective) and Orbis-DiT (continuous objective). Both use a spatiotemporal Transformer (24 layers, hidden size 768, input resolution $320 \times 640$), trained for 18 epochs with identical step counts and batch sizes; optimizer and schedule follow Mousakhan et al. (2025). Each model observes a 7-frame context (implicitly revealing the factors discussed) and autoregressively generates 30 frames one frame at a time. No clips from $\odot\rightarrow$ or $\bigcirc\leftarrow$ appear in training.

**Evaluation protocol.**  We assess compositionality via nearest-neighbor retrieval against real videos using V-JEPA2 features (Assran et al., 2025). For each generated clip, we: (i) encode frames; (ii) spatially pool to a $4 \times 6$ grid (feature dim 1024); (iii) take the last 10 generated frames and uniformly subsample 5; and (iv) flatten to a descriptor of size $4 \times 6 \times 5 \times 1024$. The real gallery contains 100 clips per composition (600 total). Distances are based on cosine similarity; we take $k = 1$ nearest neighbors and report

$$\mathrm{CRA}_k(c^\star) = \frac{1}{k} \sum_{i=1}^{k} \mathbf{1}\big\{ c(\mathrm{NN}_i) = c^\star \big\},$$

averaged over generated clips targeting composition $c^\star$. With a balanced 6-class gallery, the chance is $1/6 \approx 0.167$.

# H  LANGUAGE

To test whether our findings on continuous representations playing a role in compositional performance extend beyond the visual domain, we constructed a controlled card arithmetic dataset taking inspiration from the Points24 dataset (Chu et al., 2025). Each instance requires forming an arithmetic expression from four playing cards from a standard deck of 52 cards, such that all cards are used exactly once to reach a specified target value. To map cards to numerical values, the Ace was mapped to 1, the number cards (2-9) retained their face value, and face cards were mapped to the next integers (Jack = 10, King = 11, Queen = 12).

The data was split into compositional sets based on rules that the model had to follow while performing the arithmetic task: (i) The model could be asked to formulate an equation from the cards that satisfies

a target value of either 12 or 24, (ii) cards belonging to red suits (Hearts, Diamonds) could have the same value as the cards belonging to the black suit (Clubs, Spades), or could be double their original value. Training examples to the model contained only single-rule instances, while the test set required applying both rules simultaneously.

We tested a Llama-3.2 model (Dubey et al., 2024) for our experiments and we look at the reasoning mechansims in these models as analogous to our objectives in our vision experiments.We test the same base model under two reasoning strategies: (i) normal chain-of-thought (CoT) (Wei et al., 2023) which is the standard textual step-by-step reasoning, and (ii) COntinuous-Chain-Of-Thought (COCONUT) (Hao et al., 2024), where intermediate reasoning states are represented in a continuous latent space as "thoughts" rather than purely symbolic text.

Both models were trained on identical data and evaluated on the compositional split. Performance was measured by the proportion of trials in which the model produced a valid formula using the correct rules that satisfied the target.

We used "success rate" as our evaluation metric. Put simply, we use a statistical verifier to verify if the final output given by our model (with both standard and continuous CoT mechanisms) are correct. The success rate for the model with continuous reasoning was 12.39%, significantly higher than the standard CoT baseline, which achieved only 4.82%. Although the absolute numbers are low due to the difficulty of the task, the consistent improvement demonstrates that continuous reasoning states provide a tangible benefit for compositional generalization in language.

This small-scale experiment serves two purposes: First, it provides indications of a proof of transfer-the advantages of continuous representations we observed in *visual* generative models also hold in the language domain. Second, it highlights an early but promising signal that *continuous reasoning for language* is a fruitful research direction. While the present setup is limited and minimal, it lays the groundwork for future studies that explore richer tasks, larger models, and more principled strategies for continuous reasoning.

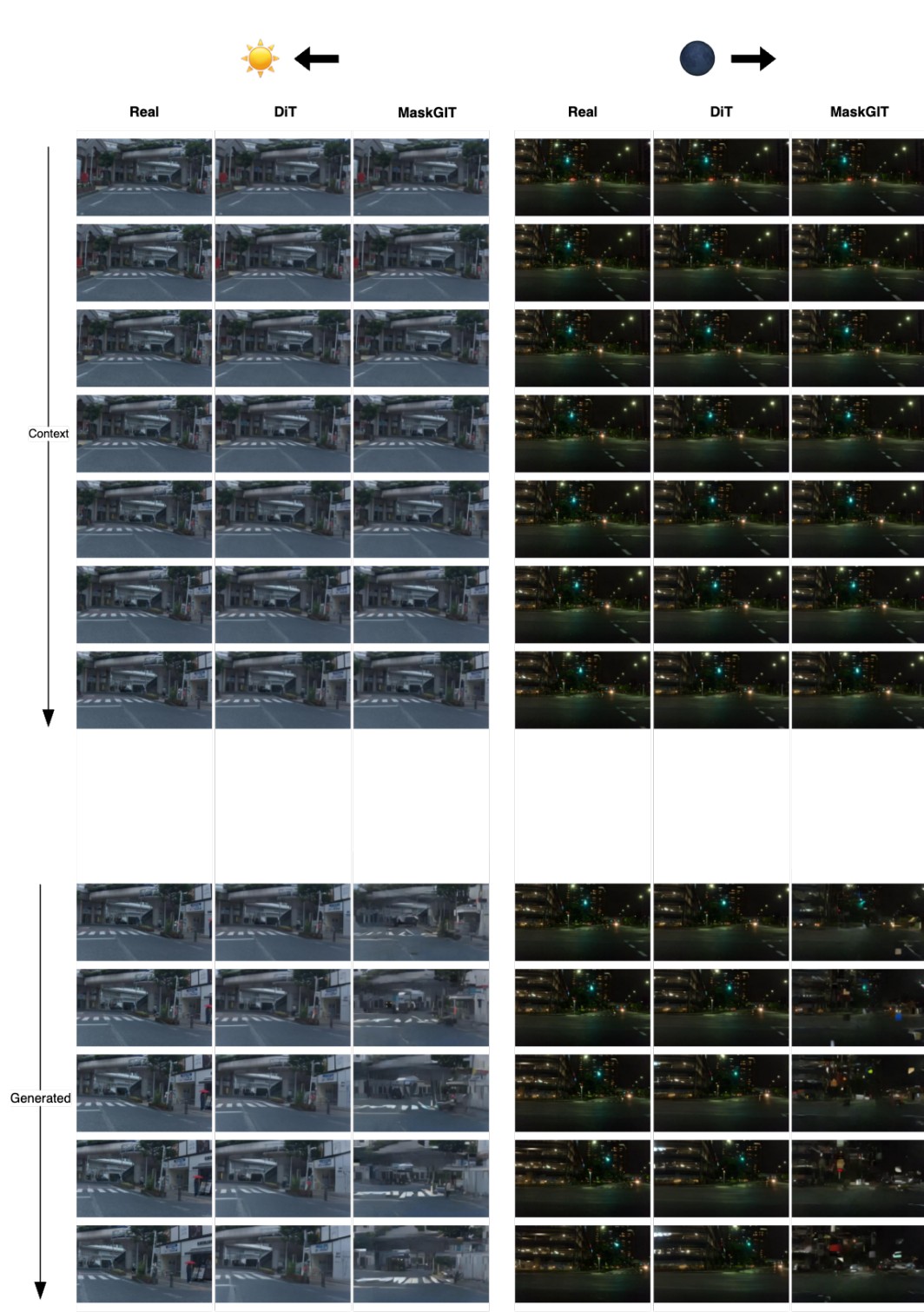

Figure 19: Qualitative results with Orbis on CoVLA. Two novel compositions are shown, ⊙→ and ○←, each conditioned on a 7-frame initial context and predicting 5 frames ahead. Orbis-DiT follows the target compositions, whereas Orbis-MaskGIT often struggles.

**System Prompt** ($v_0^{in}$)

**[Task Description]**
You are an expert in the card arithmetic game. You are observing four cards: each card must be used exactly once to form an arithmetic expression. Number cards take their face value (Ace = 1, 2–9 as given). The suits are abbreviated as Hearts = H, Diamonds = D, Clubs = C, Spades = S. Face cards are valued as J = 10, K = 11, Q = 12. In addition, if a card belongs to a red suit (Hearts H or Diamonds D), its value is doubled. Your goal is to output a formula that evaluates to the target number, which is either 12 or 24. The allowed operators are +, −, ∗, parentheses, and =.

**[Input]**
Cards: [2H, 3S, QC, 4D]
Target: 24

**[Output]**
Your response should be a valid JSON file in the following format:

```
{
  "cards": [x, y, z, w],
  "values_after_rules": [a, b, c, d],
  "target": T,
  "formula": "an equation that equals T"
}
```

---

**Model output** ($v_t^{out}$)

```
{
  "cards": ["2H", "3S", "QC", "4D"],
  "values_after_rules": [4, 3, 12, 8],
  "target": 24,
  "formula": "(12+8)+(4*3)"
}
```

---

**Verifier Output** ($v_t^{ver}$)
This trial is unsuccessful because the proposed formula does not evaluate to 24.

Figure 20: Example system prompt, unsuccessful model output, and verifier feedback for our language task with compositional rules: target restriction and red-suit value doubling.

**System Prompt** ($v_0^{in}$)

**[Task Description]**
You are an expert in the card arithmetic game. You are observing four cards: each card must be used exactly once to form an arithmetic expression. Number cards take their face value (Ace = 1, 2–9 as given). The suits are abbreviated as Hearts = H, Diamonds = D, Clubs = C, Spades = S. Face cards are valued as J = 10, K = 11, Q = 12. In addition, if a card belongs to a red suit (Hearts H or Diamonds D), its value is doubled. Your goal is to output a formula that evaluates to the target number, which is either 12 or 24. The allowed operators are +, −, ∗, parentheses, and =.

**[Input]**
Cards: [AD, KH, 5D, 9S]
Target: 12

**[Output]**
Your response should be a valid JSON file in the following format:

```
{
   "cards": [x, y, z, w],
   "values_after_rules": [a, b, c, d],
   "target": T,
   "formula": "an equation that equals T"
}
```

---

**Model output** ($v_t^{out}$)

```
{
   "cards": ["AD", "KH", "5D", "9S"],
   "values_after_rules": [1, 11, 5, 9],
   "target": 12,
   "formula": "(1+5)*(11-9)"
}
```

---

**Verifier Output** ($v_t^{ver}$)
This trial is successful because the proposed formula evaluates to 12.

Figure 21: Example system prompt, successful model output, and verifier feedback for our language task with compositional rules: no target restriction and no red-suit value doubling.

