# OpenReview forum: "What Drives Compositional Generalization in Visual Generative Models?"
_ICLR.cc/2026/Conference — Submitted to ICLR 2026_

### Official Review · Reviewer_RdpQ · 2025-10-23

**Soundness:** 3
**Presentation:** 3
**Contribution:** 2
**Rating:** 6
**Confidence:** 4

**Summary:**

This paper presents a systematic empirical investigation into the factors that enable or hinder compositional generalization in visual generative models. Through controlled experiments across multiple domains, authors demonstrate that achieving robust compositional generalization in visual generative models can be facilitated by continuous distribution modeling and full, non-quantized conditioning signals. Building on these insights, the authors propose a JEPA-based auxiliary loss that improves the compositional capabilities of MaskGIT.

**Strengths:**

1. This paper is well written.
2. To answer the central question raised by authors (``What are factors that enable or hamper compositional generalization in visual generative models?''), the authors designed extensive experiments. The results are insightful.

**Weaknesses:**

1. This paper experimentally verifies a previously intuitive factor. A deeper discussion of why discrete objectives hinder compositionality would strengthen the narrative.
2. This paper uses controlled experiments to demonstrate the relationship between continuous loss and compositional generalization. However, mainstream works such as MaskGIT or DiT are founded on large-scale pre-training, which means experiments without scaling may show completely different results. I am curious about, under the premise of large-scale pre-training, how is the generalization ability of different models for unseen combinations?
3. The result of this paper could be significant in training LLMs. However, the discussion of `Do our results extend to language?' only provides a comparison of COCONUT and CoT, which are inference-time continuous. I suggest that the comparison should be conducted in the pre-training stage to illustrate that LDM can have a higher upper limit than LM.

**Questions:**

See ``Weakness'' section.

---

> ### Author Response · Authors · 2025-11-20
>
> We thank the reviewer for their positive assessment, especially for recognizing that our experiments are extensive and that the results are insightful. We also highly appreciate that you clearly captured the central question and the purpose of the JEPA-based auxiliary loss in MaskGIT.
>
> > **W1 – “Intuitive” factor & why discrete objectives hinder compositionality.**
>
> Please refer to Reviewer drhc, W1. We will expand the discussion to explain why discrete objectives hinder compositionality in our setting: quantization ties factors to discrete, non-ordinal codebook tokens that have no notion of (semantic) distance, and induce discontinuities. For compositional generalization, however, it is crucial that factors live in a semantic space with a meaningful distance structure, as highlighted by Finding 2 (continuous representations & distance), Finding 3 (the need for detailed semantic information), and Finding 4 (controlled continuous-vs-discrete interventions & semantic disentanglement). This picture is consistent with the PAT hypothesis proposed in the concurrent work of Song et al. (2025, Sec. 5.2, Fig. 8), especially their experiment with REPA (Fig. 7(b)); our study provides an independent, complementary lens on this view and systematically explores compositional generalization across different design axes and a broader set of datasets and modalities.
>
> > **W2:** We agree that scale might influence compositional generalization.
>
> MaskGIT and DiT are typically trained on ImageNet-scale data ($\approx$1.2M images). To test our hypothesis at scale, we evaluate Orbis, an example of large-scale pretraining, trained on $\approx$43 hours of video at 5 Hz ($\approx$774,000 frames), and we observe similar patterns as with smaller models trained on synthetic data (see Table 6 and Fig. 19).  In this sense, our large-scale experiment is on par with the works cited by the reviewer; however, a more systematic study of how these properties emerge with increasing scale could merit a dedicated paper on its own. Our present work instead provides a detailed, controlled study of fundamental design choices across datasets and scales of increasing complexity (see General Comment, point (3)), which such a study would naturally build on.
>
> > **W3 – Language extension and pre-training vs inference**
>
> We agree that the language part is exploratory and not on the same footing as our visual results. However, we did go beyond pure test-time prompting by using supervised fine-tuning (SFT) (as training is deemed significant by the reviewer), which is feasible for us, whereas full pretraining is not. We will explicitly state that our results go beyond test-time towards SFT for LLMs.
>
> ---
>
> [1] Song, K., Kim, J., Chen, S., Du, Y., Kakade, S., & Sitzmann, V. (2025). Selective Underfitting in Diffusion Models. arXiv preprint arXiv:2510.01378.

---

> ### Author Response · Authors · 2025-11-27
> **Authors' Follow-up and Request for Discussion**
>
> We appreciate the reviewers' initial feedback and hope that we have provided detailed responses addressing each concern raised. As the discussion period continues, we remain eager to engage in further dialogue to clarify any remaining questions or misunderstandings about our work.
>
> We welcome any follow-up questions, requests for clarification, or suggestions for strengthening the manuscript. Constructive engagement during this phase would be invaluable for improving our work and ensuring a thorough evaluation of our contributions.
>
> Thank you for your continued consideration.

---

### Official Review · Reviewer_XRfX · 2025-10-29

**Soundness:** 2
**Presentation:** 2
**Contribution:** 2
**Rating:** 4
**Confidence:** 4

**Summary:**

This paper investigates compositional generalization in generative models. More specifically, it performs extensive experiments to ablate the recipe of DiT and MaskGIT including tokenizer, masking, objective function, output space (continuous or discrete) to find out what contributes most to compositional generalization in visual generative models. Based on the observation that continuous objective function is a key factor, the paper proposes add an auxiliary continuous JEPA-based objective term to enhance compositional performance of MaskGIT.

**Strengths:**

**Strength**
1. Understanding compositionality in generative model is an interesting direction.

2. Extensive experiments are performed to figure out the most important factors that give rise to compositional generation in the training recipes. The study is straitforward and easy to follow.

**Weaknesses:**

1. The main claim of the paper goes too generic and cannot be supported by enough evidence. Since the paper only studies a class of visual generative models such as DiT and MaskGIT, it can not claim the study of general generative models. For example, it is overstated in the title"WHAT DRIVES COMPOSITIONAL GENERALIZATION IN VISUAL GENERATIVE MODELS?". It can only support "WHAT DRIVES COMPOSITIONAL GENERALIZATION IN DiT?" and "IMPROVED TECHNIQUES for ENHANCING COMPOSITIONALITY of MaskGIT". Likewise, there are many such overstatements in the paper.

2. In section 6.2, it studies how the incorporation of an auxiliary loss enhances compositional generation performance of MaskGIT. However, it is unclear if this extra objective term degrades image generation performance. In other works, are you trading image quality for compositionality? No qualitative or quantitative (FID) results are provided. Actually, FID should be reported for all comparisons because generation quality are as important as compsitionality in generation tasks.

3. As an experimental work that simply compare models and training factors, more complicated dataset are expected. Regarding this point, opinions from other reviewrs will be taken into account.

**Questions:**

See above.

---

> ### Author Response · Authors · 2025-11-20
>
> We thank the reviewer for their insightful comments and are glad that they found our paper straightforward and easy to follow.
>
> > **W1:** The main claims are too generic and cannot be supported by enough evidence
>
> While our study does not cover all architectures, our study systematically covers fundamental design choices in modern visual generative models, not just DiT and MaskGIT. Our experimental design spans differences in representation spaces, training techniques, and conditioning mechanisms. Our tested models represent the corners of this design space, while the proposed MaskGIT+JEPA is an interpolation between the discrete and continuous representation spaces, further reinforcing our findings. We refer the reviewer to point (3) of the General Comment.
>
> > **W2:** Does the auxiliary JEPA loss degrade image generation performance? Are you trading quality for compositionality?
>
>  We refer the reviewer to point (2) of the General Comment.
>
> The reviewer claims that “FID should be reported for all comparisons because generation quality is as important as compositionality.” While we disagree on the level at which that image quality matters, we address this in three ways. (1) We now report FVD, FID, and FDD (in dinov2 space) for real datasets (CoVLA and CelebA) comparing real vs. generated samples on both the seen and novel splits; these results are consistent with our probe evaluations,  CRA-based findings, and qualitative results, and do not change any conclusions. (2) However, FID has important limitations in our setting: it is designed for matching a data distribution via a Gaussian approximation in the feature space of the inception network (trained on real data). FID is sample-inefficient (requires lots of samples) and becomes noisy in our synthetic benchmarks, where the marginal “real” distribution does not faithfully capture the relevant structure (combinations of attributes). As a result, FID, at best, conflates low-level realism with compositional correctness.  (3) This is why we use CRA as our primary metric in Tables 3, 4, and 6. CRA is a precision metric, inspired by the standard precision for generative models [1], but tailored to compositional generalization: it directly measures how often generated samples satisfy the target composition, which is precisely what our work studies.
>
> > **W3:** Please refer to point (2) of the General Comment.
>
> **CelebA**
>
> |         | UNSeen FID  | UNSeen FDD   | Seen FID    | Seen FDD       |
> |---------|-------------|-------------|-------------|----------------|
> | DiT     | **118.60**  | 993.92      | **109.96**  | 681.78         |
> | MG      | 162.38      | 1470.78     | 113.94      | 719.95       |
> | MG+JEPA | 120.76      | **980.92**  | 117.35      | **621.43**  |
>
>
> **ORBIS**
>
> |     | FVD (SEEN) | FVD (UNSEEN) |
> |-----|------------|--------------|
> | DiT | **730.3**  | **2257.51**  |
> | MG  | 2320.95    | 3197.74      |
>
> ---
>
> [1] Kynkäänniemi, T., Karras, T., Laine, S., Lehtinen, J., & Aila, T. (2019). Improved precision and recall metric for assessing generative models. Advances in neural information processing systems, 32.

---

> > ### Comment · Reviewer_XRfX · 2025-11-27
> > **Thank you for the experiments.**
> >
> > Part of my concerns are addressed, and I raised my score to 6. I still feel some claims should be made narrower.

---

> ### Author Response · Authors · 2025-11-27
> **Authors' Follow-up and Request for Discussion**
>
> We appreciate the reviewers' initial feedback and hope that we have provided detailed responses addressing each concern raised. As the discussion period continues, we remain eager to engage in further dialogue to clarify any remaining questions or misunderstandings about our work.
>
> We welcome any follow-up questions, requests for clarification, or suggestions for strengthening the manuscript. Constructive engagement during this phase would be invaluable for improving our work and ensuring a thorough evaluation of our contributions.
>
> Thank you for your continued consideration.

---

### Official Review · Reviewer_KtzG · 2025-10-31

**Soundness:** 2
**Presentation:** 2
**Contribution:** 2
**Rating:** 2
**Confidence:** 4

**Summary:**

The authors studied the design choices of visual generative models that affect models' compositionality, driven by the performance of existing models. The authors focused on 4 design choices: the tokenizer type, the underlying generative process, and the conditioning quality. The authors concluded that the tokenizer does not matter but continuous generative distributions and good conditioning quality are important. The authors also proposed to augment MaskGIT with the JEPA objective based on their analysis.

**Strengths:**

1. The paper is easy to follow.
2. The authors did a thorough analysis of DiT and MaskGIT models which helps understand the compositional generation task.
3. Experimental details are provided in the appendix.

**Weaknesses:**

1. The authors' conclusions are too strong given that the analysis is entirely based on a set of four models. Comparing these models is insufficient to conclude that using discrete distributions is not suitable for compositional generation, simply because MaskGIT underperforms. The other conclusions are also not convincing for the same reason.
2. There is no clear evidence showing that the proposed MaskGIT with JEPA loss outperforms existing models, especially when compared with DiT. Experiments on the real dataset show no visual improvement in Figure 18 and quantitative comparisons are lacking.
3. There are typos here and there. For example, the "replaces" in line 150.

**Questions:**

1. Many works composed diffusion models or energy-based models directly in the pixel space for compositional generation. How do those methods compare against the models discussed here?
2. For a practical perspective, the generation quality presented in the paper is far behind the SOTA LLM/VLM. How do they perform on these tasks?

---

> ### Author Response · Authors · 2025-11-20
>
> We thank the reviewer. We start with W2 because it is a dependency for W1.
>
> > **W2:** No clear evidence that MaskGIT with JEPA outperforms existing models
>
> This experiment was not designed to outperform DiT but to provide a test of our hypothesis about discrete representations. By adding the JEPA loss, we make MaskGIT more continuous while keeping everything else identical. This is a controlled intervention to test whether the discreteness bottleneck is the limiting factor. The comparison should therefore be between MaskGIT+JEPA vs MaskGIT and not MaskGIT+JEPA vs DiT. We ask the reviewer to re-examine Fig. 18, and we also show quantitative CRA on CelebA (Appendix E.2, Table 4) and FID (in response to Reviewer XRfX).
>
> > **W1:** Limited model selection undermines the generalizability (4 models → can’t claim “discrete is not suitable”; therefore, “other conclusions are also not convincing for the same reason”.)
>
> > “... these models is insufficient to conclude that using discrete distributions is not suitable for compositional generation, simply because MaskGIT underperforms.” + W2
>
> We refer the reviewer to the general comment (3), response to W1, and S1,S2,S3 in drhc’s comment. Our analysis of discrete models does not stop at observing that MaskGIT underperforms. The JEPA experiment is designed to address the possibility that “MaskGIT is just underperforming” for a single reason. By intervening on the objective of the same model and observing a consistent improvement in compositional generalization, we obtain evidence that the discrete objective/representation is a key limiting factor, rather than an incidental weakness of a particular model.
>
> > “The other conclusions are also not convincing for the same reason.”
>
> We believe there is a misunderstanding in how our four conclusions are grouped.  The “only four models” concern primarily applies to the continuous–discrete finding (addressed in earlier paragraph), but our other findings do not hinge on comparing the four. Our study is designed as a controlled ablation of fundamental design axes, not a broad model zoo. We isolate core choices—representation space (continuous vs. discrete), conditioning, tokenizer, and generation mechanism—and each finding is supported by a controlled comparison (e.g., using different tokenizers with DiT). Generalizability comes from such isolation, not from adding more confounding choices (e.g., improved VQ codebooks). We welcome suggestions for controlled discrete interventions that cannot simply be ported to the continuous models without reintroducing the eliminated confounders.
>
> > **Q1:** How do pixel-space compositional methods compare to the models discussed here?
>
> We thank the reviewer for this question. We clarify three important points:
> 1. We are not sure how this question can serve the findings.
> 2. We do conduct experiments in pixel-space (Appendix C, Figs 7 and 8).
> 3. We note that composable diffusion models [1], as mentioned by the reviewer, can be considered an orthogonal direction as it can be added on top of any model we study. Therefore, they do not inform our core research question about fundamental choices. In addition, their “zero-shot” claims require careful interpretation. For instance, on FFHQ, concept-specific models are trained on images containing multiple, entangled factors simultaneously. Thus, the model sees these attribute combinations during training, even if not explicitly composing them. Contrasting this to our CelebA experiments, where we test on strictly held-out compositions. This is a fundamentally different task.
>
> > **Q2:** How do SOTA LLM/VLMs perform on these compositional generalization task?
>
> Our goal in this work is not to propose a SOTA, but to help understand and inform design choices for them. To our knowledge, current LLMs do not natively generate images or videos. In the same vein, the majority of VLMs are designed for vision-language tasks (eg, VQA) and output language tokens only. The few multimodal or T2I/T2V models that do generate images (eg, Transfusion, Chameleon) incorporate image generation components that we study- typically diffusion models or autoregressive visual tokenization approaches.
> In addition, most frontier multimodal models with generation capabilities are closed-source. Thus, we cannot access their model/data to evaluate compositional generalization, understand the design choices they employ, or control for confounding factors to conduct relevant controlled ablations. Even in well-studied domains like ARC-AGI [2], SOTA models show limited compositional generalization capabilities. Since compositional generalization remains an open research problem, we hope that our findings would serve as a useful foundation in informing design choices for future models.
>
> ---
>
> [1] Liu, Nan, et al. "Compositional visual generation with composable diffusion models." European conference on computer vision. Cham: Springer Nature Switzerland, 2022.

---

> ### Author Response · Authors · 2025-11-27
> **Authors' Follow-up and Request for Discussion**
>
> We appreciate the reviewers' initial feedback and hope that we have provided detailed responses addressing each concern raised. As the discussion period continues, we remain eager to engage in further dialogue to clarify any remaining questions or misunderstandings about our work.
>
> We welcome any follow-up questions, requests for clarification, or suggestions for strengthening the manuscript. Constructive engagement during this phase would be invaluable for improving our work and ensuring a thorough evaluation of our contributions.
>
> Thank you for your continued consideration.

---

### Official Review · Reviewer_drhc · 2025-10-31

**Soundness:** 2
**Presentation:** 3
**Contribution:** 2
**Rating:** 4
**Confidence:** 4

**Summary:**

This paper conducts a systematic study of compositional generalization in modern visual generative models.
Through carefully controlled experiments across several architectures (DiT, MaskGIT, MAR, GIVT), the authors identify two key factors influencing compositionality:

1. Whether the model learns a continuous vs. discrete distribution;
2. The degree to which the conditioning signal preserves complete information about underlying factors.

While the choice of tokenizer does not fundamentally alter the compositional abilities.

They further propose an auxiliary JEPA-based loss that augments discrete models like MaskGIT with a continuous predictive objective. This addition improves compositional generalization and yields more disentangled internal representations, as confirmed by mechanistic interpretability analyses (polysemanticity and circuit overlap).

**Strengths:**

1. Compositional generalization is a fundamental and increasingly important challenge for generative models. Investigating what architectural and objective factors drive it is both interesting and highly relevant to the broader ML community.
2. The authors conducted a series of experiments considering model architecture, training objective, conditioning description, and data modality, making their statements comprehensive and generalizable.
3. By incorporating a JEPA-style continuous predictive loss into MaskGIT, the authors successfully improved the compositionality of MaskGIT, which further proved continuous representation learning outperforms discrete token-based modeling in terms of compositional generalization. Moreover, the authors performed polysemanticity and circuit analysis, showing how continuous learning objective affects the model.

**Weaknesses:**

1. Though the authors conducted systematic experiments to support their findings, however the conclusions themselves are some what intuitive:
- Continuous training objective favors compositionality: continuous representations enforce the model to learn a smooth structure which permits interpolation between elements, therefore benefiting composition. It is intuitive that continuous latents supremes discrete tokens in this ability.
- Precise conditioning is critical: This one is more appearent. If the conditioning is precise and factor-complete, the model can learn to represent and recombine each factor explicitly. Otherwise, the learned representation to the training data itself could be problematic, not to mention compositionality.

2. The experiments on MaskGIT with JEPA loss showed how the polysemanticity and the neuron overlap are reduced, however, the causality between them and compositionality is unclear.

3. Several sections still lack sufficient detail, and certain descriptions are ambiguous and confusing. For example, how are the signals binarized in Section 5? Line 404~405 said "A head is considered polysemantic if the feature similarity difference exceeds a threshold", however, the head should be considerd polysemantic if the difference is large **across multiple factor pairs**. The "Overlap(%)" in Fig.6(c) also seemed to be a mistake.

4. The paper assumes that GIVT and MAR can serve as clean “interpolations” between DiT (continuous diffusion) and MaskGIT (discrete masked modeling). However, in practice, GIVT and MAR models differ MaskGIT and DiT in more than just the objective function or masking mechanism. The experiments is not strictly rigorous in the fundamental model settings.

**Questions:**

1. Why the compositional accuracy falls low or even down to zero during the later period of training? Especially in Fig. 4(b), the line "101" achieved almost 100% accuracy in the middle of training and then falls to zero. How were the labels dropped out, and why it caused such performance？

2. How does the reduced polysemanticity and neuron overlap lead to better compositionality? If this is intuitively considered, it could also be intuitively found out that continuous training objective and precise conditioning lead to better compositionality. In other words, could the improvements stem from smoother optimization dynamics rather than semantic disentanglement?

---

> ### Author Response · Authors · 2025-11-20
>
> We thank the reviewer for their comments. We appreciate recognizing the importance of compositional generalization, the breadth of our experimental design, and the generality of our conclusions, as well as the role of the JEPA-style continuous loss in improving MaskGIT’s compositionality.
>
> >**W1:** Inuitive findings
>
> We agree, both are intuitive, but the motivation for this work is that the field has not treated these choices as settled facts.
>
> - **W1.a (continuous vs discrete).** Recent visual generative models, like Genie [1] and Emu3 [2], still adopt discrete design, and benchmarks overwhelmingly focus on visual fidelity, not on controlled compositional OOD generalization. If the “continuous > discrete for compositionality” story were settled, we would expect the model and evaluation design to reflect consistently.
>
> - **W1.b (precise conditioning).** Can generative models without access to the full data-generating factors generalize compositionally? Our results show not only how to condition, but that access to data-generating factors is essential for continuous models to generalize compositionally. Using expressive text partially mitigates this, but the issue is often aggravated by widespread of condition dropout, counterintuitively known for generalization performance. Our controlled lossy-conditioning experiments make this failure mode explicit.
> We see this as the kind of situation where a paper is needed: researchers suspect these phenomena, but design choices do not align. With controlled evidence, we aim to make this intuition actionable and to shift focus from pure visual quality to the more fundamental ML question of generalization.
>
> >**W2:** Unclear causality between JEPA intervention and masking and compositionality
>
> Our intuition behind introducing polysemanticity analysis was strictly as a diagnostic tool for understanding representational differences between models, revealing systemic patterns- architectures with more continuous disentangled representations and stronger compositional generalization.  We will expand our discussion to more clearly articulate how representational structure relates to compositional capability.
>
> >**W3:** Several sections still lack sufficient detail, and certain descriptions are ambiguous and confusing.
>
> We will try to clarify each point and address it in the paper, where necessary. For quantization of signals in Sec 5, we refer to the answer for Q1. For the polysemanticity of attention heads- we implied the key indicator being showing large differences across multiple factors, not just one. We would change our wording to “if the feature similarity difference exceeds a threshold" should be clarified to: "if the feature similarity difference exceeds a threshold for multiple unrelated factors (e.g., both color and shape), suggesting the head attends to entangled, unrelated features.” If there are any further clarity/readability issues, we would kindly ask the reviewer to point them out.
>
> > **W4:** Please see General Comment (1) “Experimental Rigor”.
>
> > **Q1.a:**  “Why the compositional accuracy falls low or even down to zero during the later period of training? …  and why it caused such performance”
>
> This was one of the surprising findings; counterintuitively, dropout causes overfitting and falling back to seen compositions (Finding 3). See Figure 11 for the qualitative results and comments on the “101” class. The model needs access to the data generating factors, which is infeasible in practice, to learn the data generating process.
>
> > **Q1.b:** “How were the labels dropped out”
>
> We refer to line 301:303 in our paper
>
> “For quantized signals, we convert continuous signals into discrete binary signals. For lossy signals, we randomly drop each factor with a 10% probability, so that, on average, one in ten factors is missing.”
>
> >**Q2:**  “Could the improvements stem from smoother optimization dynamics rather than semantic disentanglement?”
>
> We agree there is an intuitive connection, highlighted by our findings, between continuous objectives and better compositionality, and our polysemanticity analysis provides further mechanistic insight into this intuition. It suggests that smoother optimization, semantic disentanglement, and improved compositionality are likely part of the same causal chain rather than competing explanations. It reveals how continuous objectives improve compositionality and produce less entangled representations where individual factors can be independently controlled. We will clarify this in the paper and explicitly frame a more detailed causal analysis of these factors as an interesting direction for future work on elucidating such causal chain.
>
> ---
>
> [1] Bruce, J., Dennis, ... & Rocktäschel, T. (2024). Genie: Generative interactive environments. In ICML
>
> [2] Wang, X., Zhang, X., ... & Wang, Z. (2024). Emu3: Next-token prediction is all you need. arXiv preprint arXiv:2409.18869.

---

> ### Author Response · Authors · 2025-11-27
> **Authors' Follow-up and Request for Discussion**
>
> We appreciate the reviewers' initial feedback and hope that we have provided detailed responses addressing each concern raised. As the discussion period continues, we remain eager to engage in further dialogue to clarify any remaining questions or misunderstandings about our work.
>
> We welcome any follow-up questions, requests for clarification, or suggestions for strengthening the manuscript. Constructive engagement during this phase would be invaluable for improving our work and ensuring a thorough evaluation of our contributions.
>
> Thank you for your continued consideration.

---

### Author Response · Authors · 2025-11-20
**On Mischaracterizations, Experimental Rigor, and Evidence**

We thank the reviewers for their time. We are encouraged that some find the topic important, the experiments extensive, and several results insightful. At the same time, a few key criticisms that affected the scores appear to stem from misunderstandings or inaccurate characterizations  (e.g., *“The experiments is not strictly rigorous ...”*, *“No qualitative or quantitative (FID) results are provided”*).

**(1) Experimental Rigor**
> “The experiments is not strictly rigorous”
 “GIVT and MAR models differ MaskGIT and DiT in more than just the objective function or masking mechanism.”

This concern is significant; however, our experimental design is controlled as defined in Fig. 3, lines 242-245 and 250-260, all implemented models share the same backbone, tokenizer, and they differ only along the axes we study without confounders. If you have specific potential confounders in mind, could you please indicate them?

**(2) Quantitative and Qualitative Evidence**
> “No qualitative or quantitative (FID) results are provided.”
 “quantitative comparisons are lacking.”
“more complicated dataset are expected.”

These statements are inaccurate. In addition to the probing graphs, for each regime, we provide both qualitative generations and quantitative metrics:

- Synthetic 2D images (8 compositions): qual. in Fig. 15.
- Synthetic 3D images (240 compositions): qual. in Figs. 16–17 and CRA in Table 3.
- Real faces (8 compositions, including OOD): qual. in Fig. 18 and CRA in Table 4.
- Synthetic videos (CLEVRER-Kubric): quant. in Figs. 8–10 and JEPA effects on polysemanticity/probes in Fig. 12 and Table 2.
- Real driving videos (ORBIS): CRA in Table 6, and qual. in Fig. 19.
- Compositional generalization in the language domain on the Points24 dataset.

We propose CRA, based on the standard precision, tailored to compositional generalization (joint visual quality and factor composition). We will, for non-synthetic data, report FID to consolidate the CRA results without sacrificing visual fidelity (see FID results as a reply to Reviewer XRfX).
To our knowledge, there is no prior work that performs a comparably broad and controlled compositional study across such a variety of datasets.

**(3) Scope of Claims**

> “The main claim of the paper goes too generic… in the title’” “The authors' conclusions are too strong ... ” “It can only support "WHAT DRIVES COMPOSITIONAL GENERALIZATION IN DiT?" and "IMPROVED TECHNIQUES for ENHANCING COMPOSITIONALITY of MaskGIT". “

Our goal is to test fundamental design choices under controlled conditions, not to exhaust every possible architecture. We therefore focus on a small set of axes that underlie most modern visual generative models. These axes are tokenizer,  continuous vs discrete, spatial autoregression vs diffusion on different modalities. From the literature, we choose the simplest, minimally confounded instance for these axes- DiT, MaskGIT, MAR, and GIVT- rather than variants whose “improvements” rely on extra components that could also be bolted onto the other models. We touch upon language and a SOTA driving world model. Thus, our claims apply to the classes of visual generative models instantiated along these axes, not to all conceivable models, but broader than "WHAT DRIVES COMPOSITIONAL GENERALIZATION IN DiT?" and "IMPROVED TECHNIQUES for ENHANCING COMPOSITIONALITY of MaskGIT". Each would capture only one aspect of the work. We would be grateful for suggestions for an alternative that does not undermine the work.

**(4) JEPA Intervention and Comparison to DiT**

>“There is no clear evidence showing that the proposed MaskGIT with JEPA loss outperforms existing models, especially when compared with DiT.”

We do not claim that MaskGIT+JEPA outperforms DiT. The JEPA experiment is designed as a controlled intervention on the training objective of a discrete model: by adding a continuous loss while keeping everything else fixed, we observe improvements. As stated in the abstract and section 6.2, MaskGIT+JEPA is restricted to (i) improvements over MaskGIT in the novel regimes (seen in Fig. 18), and (ii) support for the continuous-vs-discrete finding highlighted by drhc as a strength.

**(5) SOTA LLM/VLM Comparisons**
> “ the generation quality presented in the paper is far behind the SOTA LLM/VLM. How do they perform on these tasks?”

Our focus is on visual generative models, not on models with either language input space or output space, and we are not aware of SOTA LLM/VLM being evaluated on comparable controlled visual compositionality benchmarks (seen vs unseen factor splits). Existing works typically focus on visual quality not generalization. Regarding compositional generalization, most SOTA LLM/VLMs struggle with complex compositional generalization, e.g., ARC-AGI-2. Regarding SOTA, our experiment on Orbis, further validates Finding 2.  If you have an LLM/VLM that preserves control and fits the scope, please share.

---

### Author Response · Authors · 2025-11-30

We would like to thank the AC and reviewers for the effort being put into navigating this difficult situation. However, we are concerned that, after a careful, detailed rebuttal, and responses to inaccurate characterizations of the work that negatively affected some of the scores, the only change being reverted is the part that reflects the one reviewer who actually engaged with the discussion.

Some of the initial reviews were significantly influenced by mischaracterizations of our work, which we addressed in detail both in the general comment “On Mischaracterizations, Experimental Rigor, and Evidence” and in our point-by-point responses. Before the discussion period was cut short, only one reviewer had the opportunity to update their assessment in light of these clarifications, and did so positively. Reverting to the pre-discussion scores preserves the noise of those initial misunderstandings and removes the visible evidence that the dialogue meaningfully changed at least one reviewer’s view (and potentially would have changed others as well, had the discussion continued).

We respectfully ask that, in the meta-review, our rebuttal, which directly addresses the concerns and mischaracterizations of the work, and the ensuing exchange be taken into consideration and weighed accordingly, given the current situation.

---

### Meta-Review · Area_Chair_SJ6M · 2026-01-06

**Summary:**

Reviewers expressed concerns that the paper overstates its claims given the limited scope of the empirical evaluation. In particular, multiple reviewers noted that conclusions are drawn from comparisons across only four model variants, which is insufficient to support broad claims, and that the proposed method does not consistently demonstrate clear visual improvements (KtzG, Xrfx).

Several reviewers also highlighted a lack of in-depth analysis. Some results were described as largely intuitive, and the paper does not sufficiently explain the underlying reasons for the observed behaviors or provide deeper insight into why discretization may hinder learning and generalization (drhc, RdpQ).

Although the authors added quantitative metrics such as FID scores and included additional experiments on larger pretrained models in response to reviewer feedback (Xrfx, RdpQ), these additions were not sufficient to fully address concerns regarding the strength of the empirical evidence, the generality of the conclusions, and the depth of the analysis. As a result, reviewers remained unconvinced that the paper’s claims are adequately supported.

**Reviewer Concerns:**

### Addressed

1. **Quantitative evaluation:**
   The authors added quantitative FID results to strengthen the empirical evaluation (Xrfx).

### Outstanding

1. **Overstated claims based on limited model comparisons:**
   Multiple reviewers noted that the paper draws broad conclusions based on experiments with only four sets of models, which may not be sufficient to support the claimed generality (KtzG, Xrfx). Addressing this concern would require major revisions, such as substantially broader empirical validation or a significant narrowing of the claims.

2. **Lack of in-depth analysis supporting the conclusions:**
   Reviewers highlighted that the paper lacks a deep analysis explaining why discretization hinders compositionality. While the authors argue that discrete representations “live in a semantic space with a meaningful distance structure,” the current empirical evidence is not sufficient to substantiate this claim (drhc, RdpQ). Deeper theoretical justification or more thorough empirical analysis is needed to support the conclusions.

**Reviewer Scores:**

Reviewer **drhc** raised concerns that the results are largely intuitive and that the paper lacks in-depth analysis. While the authors clarified their motivation, the response does not substantially strengthen the depth of analysis. As a result, this reviewer is likely to remain negative.

Reviewer **KtzG** expressed concern that the paper overstates its claims, given that only four model variants are compared and that no clear visual improvements are observed for the proposed method. The authors highlighted qualitative results and emphasized quantitative improvements, and further argued that the four model variants are essential for the study. However, without major revisions to the claims, this reviewer is likely to remain negative.

Reviewer **Xrfx** similarly questioned whether comparisons across only four models are sufficient to support the paper’s claims and requested stronger quantitative evidence, particularly FID scores. The authors added FID results in response. This reviewer is likely to maintain their current score or potentially raise it.

Reviewer **RdpQ** raised concerns about the lack of a deep discussion on why discretization hinders learning and questioned generalization to large pretrained models. The authors provided further explanations and added experiments on the Orbis model. This response is likely sufficient for the reviewer to remain positive.

---

### Decision · Program_Chairs · 2026-01-26

Reject